# Myelination of parvalbumin interneurons shapes the function of cortical sensory inhibitory circuits

Najate Benamer [1✉], Marie Vidal [1], Maddalena Balia[1,3] & María Cecilia Angulo [1,2✉]

Myelination of projection neurons by oligodendrocytes is key to optimize action potential conduction over long distances. However, a large fraction of myelin enwraps the axons of parvalbumin-positive fast-spiking interneurons (FSI), exclusively involved in local cortical circuits. Whether FSI myelination contributes to the fine-tuning of intracortical networks is unknown. Here we demonstrate that FSI myelination is required for the establishment and maintenance of the powerful FSI-mediated feedforward inhibition of cortical sensory circuits. The disruption of GABAergic synaptic signaling of oligodendrocyte precursor cells prior to myelination onset resulted in severe FSI myelination defects characterized by longer inter-nodes and nodes, aberrant myelination of branch points and proximal axon malformation. Consequently, high-frequency FSI discharges as well as FSI-dependent postsynaptic latencies and strengths of excitatory neurons were reduced. These dysfunctions generated a strong excitation-inhibition imbalance that correlated with whisker-dependent texture discrimination impairments. FSI myelination is therefore critical for the function of mature cortical inhibitory circuits.

[1] Université de Paris, Institute of Psychiatry and Neuroscience of Paris (IPNP), INSERM U1266, "Team Interactions between neurons and oligodendroglia in myelination and myelin repair", F-75014 Paris, France. [2] GHU PARIS psychiatrie & neurosciences, F-75014 Paris, France. [3] Present address: Institut des Maladies Neurodégénératives, CNRS UMR 5293, Université de Bordeaux, Centre Broca Nouvelle-Aquitaine, F-33076 Bordeaux, France. ✉email: najate.benamer@parisdescartes.fr; maria-cecilia.angulo@parisdescartes.fr

In the central nervous system (CNS) of vertebrates, myelination of neuronal fibers by oligodendrocytes (OLs) increases the conduction and precision of action potential propagation. Experimental and modeling analyses led to the conclusion that myelination of projection neurons synchronizes remote brain areas over remarkably long distances and refines the computational power of complex neuronal networks[1–4]. Proper myelination is therefore key in shaping CNS functions. However, although first described in different species during the 80s, the myelination of local GABAergic interneurons did not attract attention for several decades[5]. Only very recently, reports demonstrated that a large fraction of the myelin present in the neocortex and hippocampus—sometimes more than 50%—enwraps the axons of GABAergic interneurons, mainly of parvalbumin (PV)-expressing cells[6–10]. Although the physiology of this important interneuron subtype has been a subject of intense investigation during the last thirty years[11], the role played by PV interneuron myelination in regulating the properties and function of these cells remains unknown. Whether myelination of short-range projection neurons, whose axons are confined to a limited space, is relevant for the function of local circuits is also elusive.

In the neocortex, the temporal and spatial precision of neuronal networks highly depends on the activity of PV+ fast-spiking interneurons (FSI). In the mouse barrel cortex, sensory inputs recruit a fast and robust PV+ FSI-mediated inhibition, particularly in layer IV, that restricts the response of excitatory neurons favoring accurate sensory processing during whisker-related behaviors[12]. It is assumed that the fast and precise synaptic signaling of PV+ FSI in neuronal circuits relies on multiple specialized intrinsic properties of these cells[11]. Notably, the reliable and high-speed propagation of action potentials of PV+ FSI depends on the high levels of Na+ channel expression in the axon and the initiation of the discharge near the soma[13]. Furthermore, PV+ FSI densely innervate a large number of target neurons at perisomatic domains providing a powerful inhibition[11]. Whether the myelination of PV+ FSI, i.e. an extrinsic rather than an intrinsic factor, could influence the fast-spiking phenotype and temporal precision of these cells in local networks has yet to be shown.

In addition to their critical role in cortical inhibition, PV+ FSI represents the preferential synaptic input onto oligodendrocyte precursor cells (OPCs) during early postnatal development[8,14]. Prior to the onset of cortical myelination, PV+ FSI neurons target perisomatic postsynaptic sites of OPCs that express GABA_A receptors containing γ2 subunits, while other interneurons contact more distal sites lacking this subunit[14]. Despite controversial results on the role of neuron-OPC synapses[15,16], the γ2-mediated GABAergic synaptic signaling of OPCs does not seem to regulate OPC development onto oligodendrocytes at a period of high proliferation and differentiation of these progenitors, i.e. the second postnatal week of the mouse[6,17]. Moreover, the deletion of the γ2 subunit in OPCs does not result in modifications of either the global amount of myelin or the number of myelinated PV interneurons in the neocortex[6]. Nonetheless, it has remained unclear whether PV interneuron-OPC synapses could guide the correct myelination of the presynaptic interneuron, impacting its function in neuronal circuits[15].

Here, we show that the genetic inactivation of γ2-mediated GABAergic synaptic signaling of OPCs prior to cortical myelination onset resulted in severe myelination defects of PV+ FSI at mature stages as revealed by a modified myelin distribution, longer internodes, and nodes, an aberrant myelination of branch points and an abnormal proximal axon morphology. Importantly, these myelin defects of PV+ FSI reduced their high firing frequency, delayed the predicted conduction velocity of their action potentials, and decreased their connectivity with excitatory neurons. Consequently, FSI-mediated feedforward inhibition is reduced in cortical sensory circuits, causing a strong excitation/inhibition imbalance. Furthermore, these morphological and physiological dysfunctions of PV+ FSI were accompanied by an impaired whisker-dependent texture discrimination behavior. Collectively, our findings reveal that myelination of PV+ FSI is required to ensure the morphological and functional integrity of their axon as well as the prominent synaptic connectivity and inhibitory role of these interneurons in local neuronal networks. We also concluded that PV+ FSI-oligodendroglia interactions during postnatal development constitute a key step in the construction of cortical circuits involved in rapid and strong inhibition during sensory processing.

## Results

**Disruption of FSI–OPC interactions causes axonal and myelination defects in FSI.** Early in postnatal development, cortical PV+ FSI transiently contacts OPCs at postsynaptic sites containing the γ2 subunit of GABA_A receptors (γ2-GABA_AR)[14,18]. As described in layer V of the barrel cortex[6], the inactivation of these synapses in *NG2^creERT2;Gcamp3;γ2* (γ2^f/f) mice caused a significant decrease in the amplitude of GABAergic postsynaptic currents (PSCs) recorded in layer IV OPCs in response to thalamic stimulation without affecting the amplitude of glutamatergic PSCs (Supplementary Fig. 1a–e; see Online Methods). This disruption of γ2-GABA_AR-mediated synaptic activity in OPCs did not modify the densities of recombinant Olig2-expressing OL lineage cells or OPCs at the peak of FSI–OPC connectivity, *i.e.* postnatal day 10 (P10)[6,8,14] (Supplementary Fig. 1f–i).

Since PV+ FSI represent the largest proportion of cortical myelinated interneurons and constitutes the main GABAergic synaptic input of OPCs during cortical development[8,9,14], an interesting possibility would be that the early FSI–OPC communication drives the proper myelination and the correct axon maturation of these interneurons. This was tested by comparing the axon morphology and myelin distribution of single biocytin-loaded FSI, randomly recorded in control and γ2^f/f mice at P25-P30 in layer IV (Fig. 1 and Supplementary Fig. 2a, b). Indeed, layer IV FSI plays a key role in neuronal inhibition and, ultimately, in whisker-dependent object discrimination during sensory processing[12]. The general morphology of FSI axons was assessed by 3D reconstructions where only clearly interconnected branches were taken into account. We found that the reconstructed axonal length as well as the complex axonal arborization, as showed by Sholl analysis, was unchanged between control and γ2^f/f mice (Supplementary Fig. 2c; reconstructed axonal length: 3565 ± 437.7 μm for n = 8 FSI in control mice and 3321 ± 512.9 μm for n = 7 FSI in γ2^f/f mice; p = 0.7789, two-tailed Mann–Whitney U test). As recently reported[6,7,9], all 3D reconstructions of recorded FSI were myelinated in both groups as revealed by the presence of internodes expressing the myelin basic protein (MBP) which were accommodated on only ~10% of the initial region of the reconstructed axon, and thus exhibited a biased distribution towards proximal axonal segments (Fig. 1a, b and Supplementary Fig. 2d). In addition, the mean interbranch axon length of the myelinated region was also unaffected in the mutant (Supplementary Fig. 2e, f). As previously reported[9], short interbranch point segments were unmyelinated in the two groups (Supplementary Fig. 2f), suggesting that the same spatial rules govern segmental FSI myelination by OLs in control and γ2^f/f mice.

Despite the similarities in the complexity of FSI axons in the two groups, a close inspection of the myelinated segments showed that both the proximal axon morphology and myelin distribution were abnormal in the mutant. First, we found that the distance

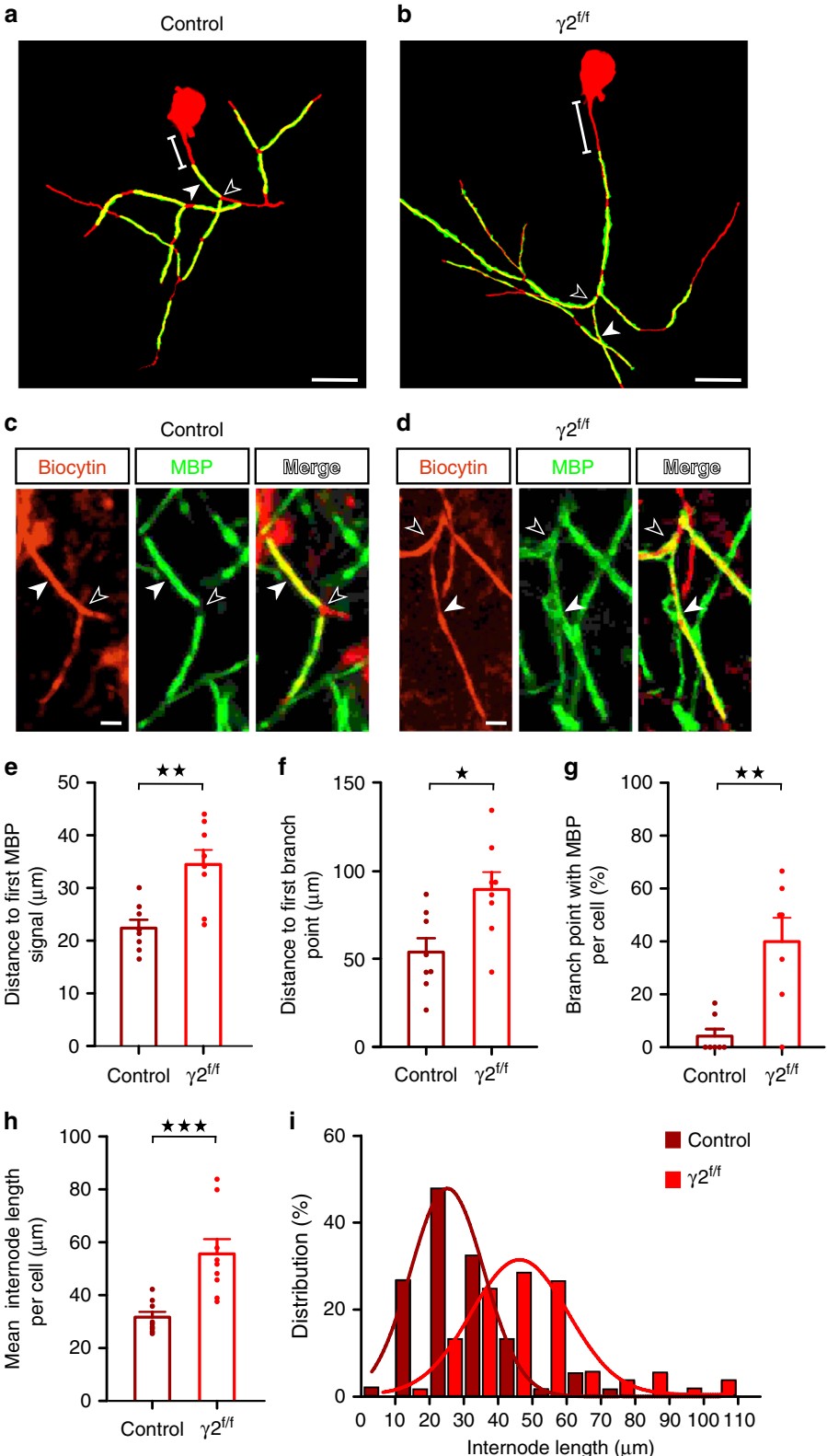

from the soma to the onset of MBP immunofluorescence signal was significantly increased in FSI from γ2$^{f/f}$ mice compared to controls (Fig. 1a, b, e). Interestingly, this displacement of the onset of proximal axon myelination was also observed in immunostained PV$^+$ interneurons of layer V where both PV$^+$ FSI and myelination are abundant[19] (Supplementary Fig. 3a, b), corroborating that most FSI axons displayed this defect in the

mutant. However, the elongation of the initial part of the axon in γ2$^{f/f}$ mice was not accompanied by an increased length or a displacement of PV$^+$/Ankyrin G$^+$ axon initial segment (AIS) (Supplementary Fig. 4). Furthermore, the axon segment from the soma to the first branch point exhibited a short distance and typically accommodated a single internode in controls[9] (Fig. 1a, f). In contrast, this proximal axonal part showed a significantly

**Fig. 1 Myelination defects of FSI in γ2$^{f/f}$ mice. a, b** Representative 3D reconstructions of myelinated FSI from control (**a**) and γ2$^{f/f}$ (**b**) mice showing biocytin (red) and MBP (green) labeling. Note that the distance from the soma to the onset of MBP signal is longer in the γ2$^{f/f}$ mouse compared to the control (lines). The first branch point (open arrowhead) and an internode (solid arrowhead) are indicated. Scale bar: 40 μm. **c, d** Confocal images from reconstructed FSI in (**a**) and (**b**) showing internodes (solid arrowheads) and axonal branch points (open arrowheads). Note the longer MBP$^+$ internode co-labeled with biocytin and the myelinated branch point in the γ2$^{f/f}$ mouse. Scale bar: 10 μm. We reconstructed a total of $n = 8$ cells from $N = 5$ control mice and $n = 9$ cells from $N = 3$ γ2$^{f/f}$ mice. **e, f** Dot plots of the distances from the soma to the first MBP signal (**e**) and from the soma to the first branch point (**f**) in control (dark red) and γ2$^{f/f}$ (light red) mice ($n = 8$ cells from $N = 5$ control mice and $n = 8$ cells from $N = 3$ γ2$^{f/f}$ mice; $p = 0.0047$ for (**e**) and $p = 0.02$ for (**f**), two-tailed Mann–Whitney U test). **g** Dot plots of the percentage of branch points with MBP per cell in control (dark red) and γ2$^{f/f}$ (light red) mice for the same cells ($n = 7$ cells from $N = 5$ control mice and $n = 7$ cells from $N = 3$ γ2$^{f/f}$ mice; $p = 0.0095$, two-tailed Mann–Whitney U test). **h** Dot plots of the mean internode length per cell for control (dark red) and γ2$^{f/f}$ (light red) mice ($n = 8$ cells from $N = 5$ control mice and $n = 9$ cells from $N = 3$ γ2$^{f/f}$ mice; $p = 0.0003$, two-tailed Mann–Whitney U test). Dot plots in (**e–h**) are presented as mean±s.e.m and dots represent data from individual FSI. **i** Distribution of internode lengths for the same cells in control and γ2$^{f/f}$ mice. Fitted lines show Gaussian distributions in control (dark red) and γ2$^{f/f}$ mice (light red) (Gaussian fit peak value: 21.68 ± 10.5 μm and 39.09 ± 13.71 μm for control and γ2$^{f/f}$ mice, respectively). Note the shift towards longer values in γ2$^{f/f}$ mice ($D = 0.4747$; $p < 0.0001$, two-sided Kolmogorov–Smirnov test).

longer distance and displayed two internodes in half of the analyzed neurons in γ2$^{f/f}$ mice (Fig. 1b, f). Moreover, the mean internode length per cell was highly increased (Fig. 1a–d, h). This myelination defect was particularly evident when comparing the internode length distributions between controls and γ2$^{f/f}$ mice, since a significant shift occurred towards longer internodes in the mutant (Fig. 1i). Notably, longer internodes were often accompanied by myelinated branch points in γ2$^{f/f}$ mice (Fig. 1c, d, g), a severe defect if we consider that these strategic axonal regions are not usually myelinated, thus ensuring the adequate propagation of action potentials[20].

In addition to internodes and branch points, nodes of Ranvier are important axonal structures whose length contributes to the optimization of action potential propagation and conduction velocity[21]. To investigate whether FSI myelination impairments were associated with changes in node length, we identified the nodes as the structure lying between two consecutive myelinated internodes flanked by two Caspr-labeled paranodes and used the plot intensity profiles of Caspr fluorescence to measure their length (Fig. 2a–c). Remarkably, we found an increased mean node length per cell accompanied by a decreased number of paranodes per μm in γ2$^{f/f}$ mice (Fig. 2d, e). In fact, the comparison of node length distributions for all cells between controls and γ2$^{f/f}$ mice exhibited a significant shift toward longer nodes (Fig. 2f). It is noteworthy that we did not find any correlation between the mean node length and the axonal length to the first MBP signal or to the first branch point in controls and γ2$^{f/f}$ mice, suggesting that the length of these different axonal subregions are most likely independently regulated (correlation coefficient of 0.250, $p = 0.595$ in controls and of 0.336, $p = 0.882$ in γ2$^{f/f}$ mice for the correlation of node length with a length to the first MBP signal; correlation coefficient of −0.498, p = 0.256 in controls and of 0.005, p = 0.991 in γ2$^{f/f}$ mice for correlation of node length with a length to the first branch point; Spearman correlation). Moreover, the Caspr protein, detected in immunostainings and normally expressed at the extremity of MBP$^+$ internodes facing branch points, was never expressed at this axonal site in abnormally myelinated ramifications of FSI in γ2$^{f/f}$ mice, suggesting that branch points aberrantly covered by myelin in the mutant might lack proteins present in nodes of Ranvier.

Severe myelination defects and axon malformation of FSI most probably resulted from altered OL lineage cells in layer IV. To test this possibility, we analyzed the morphology of biocytin-loaded recombinant cells recognized by their fluorescence during patch-clamp recordings at P26-P35. Among the recorded cells, we identified OPCs by a rectifying I–V curve, the presence of inward Na$^+$ currents and an immunoreactivity for NG2[22]. (Supplementary Fig. 5a, b). We also found that all recorded OPCs lacked the OL marker CC1 (Supplementary Fig. 5a, b). Morphometric

measurements using Sholl analysis did not reveal any difference in the morphological complexity of OPC processes between control and γ2$^{f/f}$ mice (Supplementary Fig. 5c–f). In comparison, OLs undergoing differentiation were identified by a linear I–V curve and an immunoreactivity for CC1 (Fig. 3a, b). Although we tested in four occasions that these cells lacked the OPC marker NG2, none of these cells expressed MBP, even at P84-P93, indicating that they were not fully mature in both groups (Fig. 3a–d; for MBP, $n = 12$ and $n = 11$ for $N = 6$ control and $N = 6$ γ2$^{f/f}$ mice, respectively). It is noteworthy that Gcamp3 fluorescence decreases with the maturational state of the cells in acute slices of these mice, which prevented us from detecting myelinating OLs during recordings. Nevertheless, morphometric measurements of biocytin-loaded differentiating OLs revealed that the number of branch intersections from the soma was reduced in γ2$^{f/f}$ mice compared to controls (Fig. 3c–e). We also found a significant decrease in the number of branches as well as a reduction of the mean and sum of branch crossings from the soma between the two mice (Fig. 3f–h), while the maximal distance reached by the branches remained unchanged (Fig. 3i). These data showed a lower degree of ramification without a shortening of main branches of recombinant layer IV differentiating OLs in the mutant, suggesting that these cells might be in a distinct differentiation state.

Collectively, these results show that the inactivation of γ2-GABA$_A$R-mediated synaptic activity of OPCs prior to myelination onset resulted in an altered topography of FSI myelination. While the general maturation of complex and ramified axons did not appear to be affected in FSI of mutants, myelination defects in most if not all FSI caused an abnormal proximal axon morphology, an aberrant myelination of axonal branch points and an increased length of internodes and nodes of Ranvier. Since proximal axon morphology and myelination are critical for the generation and propagation of action potentials[20], myelination defects of FSI may impact signal propagation and conduction.

**Adverse effect of abnormal myelination on the FSI spiking phenotype.** Modifications of axon architecture and myelin distribution, particularly at the proximal axon segment, are known to change the characteristics of action potential discharges[20,23]. To evaluate whether myelination defects of FSI induce alterations of their intrinsic electrophysiological properties, we evoked action potentials of recorded cells by injecting somatic currents in acute slices of P21-P30 control and γ2$^{f/f}$ mice. As previously described[24], layer IV FSI in controls were mainly characterized by narrow action potential waveforms with profound after-hyperpolarizations (AHPs) and a very high firing frequency displaying a limited accommodation during trains (Fig. 4a and Supplementary Table 1). While no changes were observed in

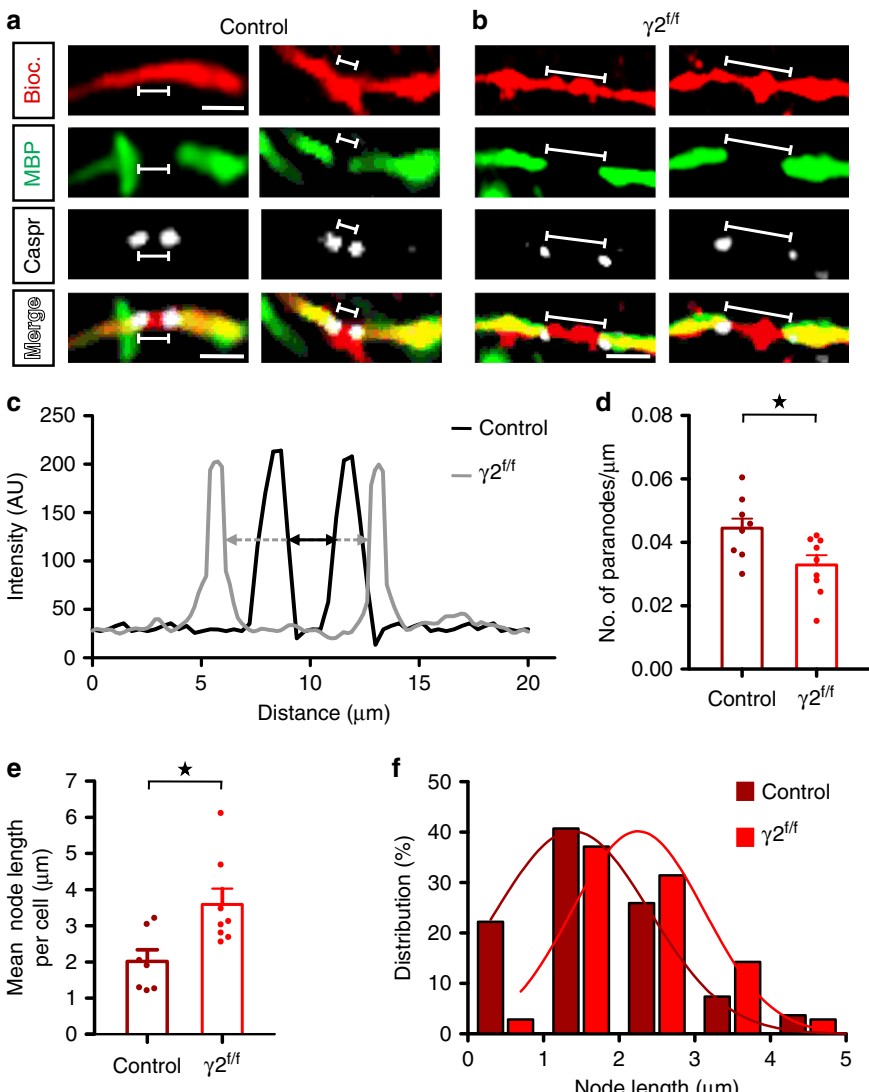

**Fig. 2 Node of Ranvier length of FSI axons is increased in γ2^{f/f} mice. a, b** Representative confocal images of single nodes of Ranvier flanked by Caspr-expressing paranodes (gray) of FSI axons labeled with biocytin (red), MBP (green) in control (**a**) and γ2^{f/f} (**b**) mice. Note that nodes of Ranvier are longer in γ2^{f/f} mice compared to controls (lines). Data of nodes of Ranvier were obtained from reconstructed FSI in Fig. 1 ($n = 8$ cells from $N = 5$ control mice and $n = 9$ cells from $N = 3$ γ2^{f/f} mice). Scale bar: 5 µm. **c** Intensity profiles of Caspr staining for the first nodes of Ranvier in control (**a**) and γ2^{f/f} (**b**) mice. Node length was measured at the half maximum intensity of each paranode (black and gray arrows). **d, e** Dot plots of mean number of paranodes per µm (**d**) and mean node length per cell (**e**) of the same FSI axons of Fig. 1 for control (dark red) and γ2^{f/f} (light red) mice ($n = 8$ cells from $N = 5$ control mice and $n = 9$ cells from $N = 3$ γ2^{f/f} mice, $p = 0.021$ for (**d**); $n = 7$ cells from $N = 5$ control mice and $n = 8$ cells from $N = 3$ γ2^{f/f} mice, $p = 0.028$ for (**e**); two-tailed Mann–Whitney U test). Dot plots in (**d**) and (**e**) are presented as mean±s.e.m and dots represent data from individual FSI. **f** Distribution of node lengths for the same cells in control and γ2^{f/f} mice. Fitted lines show Gaussian distributions in control (dark red) and γ2^{f/f} mice (light red) (Gaussian fit peak value: 1.58 ± 1 µm µm and 2.05 ± 0.87 µm for control and γ2^{f/f} mice, respectively). Note the shift towards longer values in γ2^{f/f} mice ($D = 0.3778$; $p = 0.014$, two-sided Kolmogorov–Smirnov test).

input resistance, resting potential, AHP, spike threshold, amplitude and duration, a significant decrease in the firing rate of layer IV FSI occurred in γ2^{f/f} mice (Fig. 4a, c and Supplementary Table 1). Interestingly, a similar firing defect was found in layer V PV^+ FSI (Supplementary Fig. 3c–e). To test the specificity of the relationship between FSI myelination impairments and a reduced FSI firing frequency, we also analyzed the same electrophysiological properties of recorded spiny stellate cells (SSCs), a second major neuronal component of layer IV which supports cortical columnar processing in the barrel cortex[12]. No changes in passive and active properties were observed in SSCs (Fig. 4b, d and Supplementary Table 1). Finally, we also analyzed the electrophysiological properties of layer V pyramidal neurons, a cell

type known to be highly myelinated in the cerebral cortex, and did not find any difference either (Supplementary Table 1). These data indicate that the abnormal axon morphology and myelination of FSI changed the intrinsic excitability of these interneurons but not that of SSCs or pyramidal cells.

During postnatal development, FSI undergoes important morphological, biochemical, and electrophysiological changes[14,25,26]. Notably, the low firing rate of FSI in the mutant recalls the slow, immature PV^+ FSI firing pattern described at P10, i.e. at the peak of FSI–OPC synaptic activity[8,14,27], suggesting that besides inducing an aberrant myelination and axon morphology, the inactivation of γ2-mediated synapses of OPCs at an early postnatal period compromised FSI maturation.

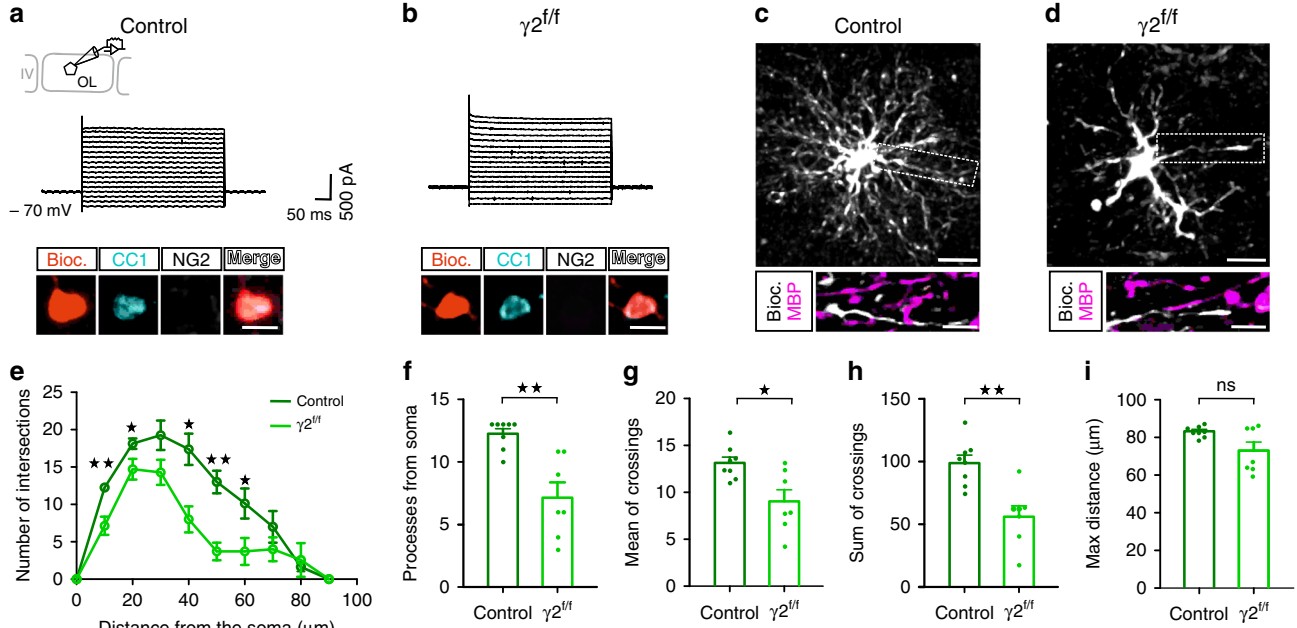

**Fig. 3 Recombinant layer IV differentiating OLs exhibited an aberrant morphological complexity in γ2^f/f mice. a, b** Currents induced by voltage steps from +60 mV to −100 mV for biocytin-loaded recombinant cells held at −70 mV, positive for the OL maker CC1 and negative for the OPC marker NG2 from control (**a**) and γ2^f/f (**b**) mice. Note the linear I-V curve of these cells in both groups. Scale bar: 10 μm. **c, d** Representative confocal images of recorded recombinant layer IV differentiating OLs loaded with biocytin (gray) from control (**c**) and γ2^f/f (**d**) mice (*n* = 8 cells from *N* = 3 control mice and *n* = 7 cells from *N* = 3 γ2^f/f mice). Note the lack of MBP co-staining in branches (insets). Scale bar: 40 and 10 μm. **e** Sholl analysis of the arborization of recorded recombinant layer IV differentiating OLs showing a decreased cell complexity between 10 and 60 μm from the soma (*n* = 8 cells from *N* = 3 control mice and *n* = 7 cells from *N* = 3 γ2^f/f mice; *p* = 0.0038, *p* = 0.0298, *p* = 0.0204, *p* = 0.0038, and *p* = 0.0362 at 10 μm, 20 μm, 40 μm, 50 μm, and 60 μm, respectively; multiple two-tailed Mann–Whitney U test). **f** Dot plots of the number of branches from soma per differentiating OL in control (dark green) and γ2^f/f (light green) mice (*n* = 8 cells from *N* = 3 control mice and *n* = 7 cells from *N* = 3 γ2^f/f mice; *p* = 0.028, two-tailed Mann–Whitney U test). **g, h** Dot plots of mean (**g**) and sum (**h**) of crossings per differentiating OLs in control (dark green) and γ2^f/f (light green) mice (*n* = 8 cells from *N* = 3 control mice and *n* = 7 cells from *N* = 3 γ2^f/f mice; *p* = 0.0289 and *p* = 0.0022 for (**e, f**), two-tailed Mann–Whitney U test). **i** Dot plots of maximum distance reached by branches per differentiating OL in control (dark green) and γ2^f/f (light green) mice (*n* = 8 cells from *N* = 3 control mice and *n* = 7 cells from *N* = 3 γ2^f/f mice; *p* = 0.3248, two-tailed Mann–Whitney U test). Dot plots in (**f-i**) are presented as mean±s.e.m and dots represent data from individual OLs.

In normal conditions, moderate levels of PV mRNAs and protein expression in FSI at P10 increase during the following three postnatal weeks in the somatosensory cortex[25,27]. To assess potential FSI developmental impairments in γ2^f/f mice, we analyzed the density of PV cells at P10, P24, P30, and P120 (Fig. 4e, f). Although we observed the expected increase in PV cells from P10 to P30 in control mice, their density remained stable during the first postnatal month in the mutant, after which it only increased at P120 (Fig. 4e, f). Neither the density of PV+ neurons nor the ratio PV+ cells/NeuN+ cells changed during adulthood, suggesting a delayed expression of the PV protein rather than a loss of PV cells in γ2^f/f mice (Fig. 4f and Supplementary Fig. 6). In addition, while no changes were detected at P10 and P120 between control and γ2^f/f mice, significant differences were observed at P24 and P30, indicating that the early disruption of FSI–OPC interactions resulted in a PV expression deficiency during the developmental myelination process (Fig. 4f).

Altogether, these observations demonstrate that FSI myelination defects selectively hamper the high-frequency firing of these interneurons. Since the OPC synaptic impairment occurs prior to myelination onset and induces PV expression deficits in γ2^f/f mice, it also seems that morphological and functional FSI defects result from an abnormal developmental maturation of these interneurons in the mutant. These results point to an important role of γ2-mediated synapses of OPCs in FSI development, raising

the interesting possibility that FSI-oligodendroglia interactions influence the functional maturation of FSI in inhibitory circuits.

**Deterioration of FSI-mediated feedforward inhibition and conduction velocity.** In the mouse barrel cortex, sensory inputs from the ventral posterior medial nucleus of the thalamus (VPM) project more prominently within layer IV to directly contact both inhibitory FSI and excitatory SSCs (Fig. 5a). The fast integration of FSI-mediated inhibition in this layer provides a robust and highly specific feedforward inhibition to SSCs[12,28]. Mature FSI-SSC circuits result from the coordinated increase in FSI-SSC synaptic connectivity and thalamic drive onto FSI starting from the second postnatal week[29], i.e. at the peak of FSI–OPC connectivity[8,14]. We therefore tested whether FSI myelination impairments impact the maturation and function of layer IV FSI-SSC circuits.

In the normal cortex, thalamic fiber stimulation elicited small direct excitatory postsynaptic currents (EPSCs) in SSCs with respect to larger disynaptic FSI-mediated inhibitory postsynaptic currents (IPSCs) in young and mature circuits of acute thalamocortical slices[28,29] (Fig. 5a, b). No significant changes in EPSC amplitudes were observed at different developmental stages in both groups, indicating that the excitatory thalamocortical input did not change during development and was not affected in γ2^f/f mice (Fig. 5a–c). In contrast, we observed a significant

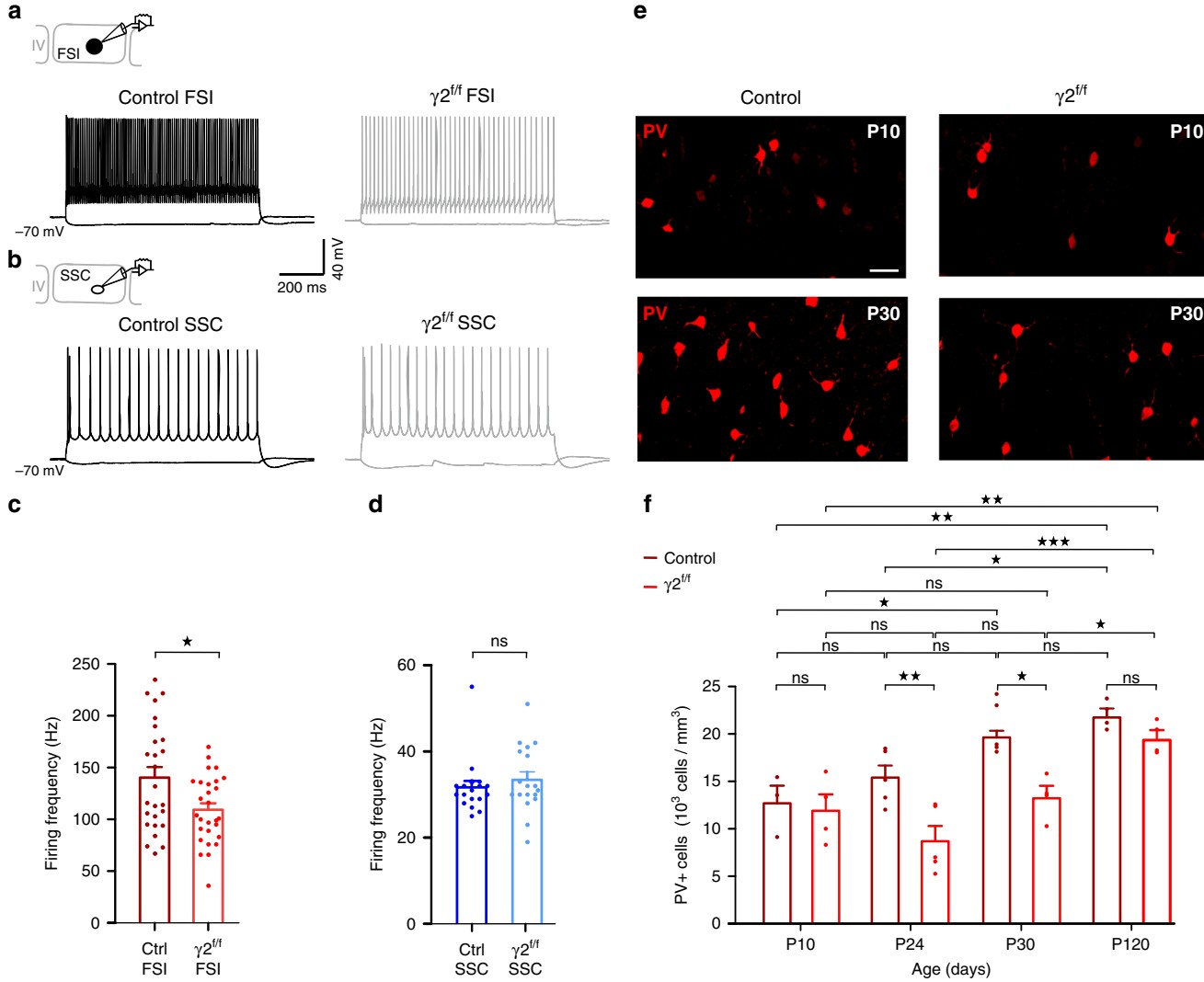

**Fig. 4 FSI firing frequency and PV expression maturation in control and γ2^f/f mice. a, b** Current clamp recordings of FSI (**a**) and SSCs (**b**) held at −70 mV during injections of 200 pA and −50 pA in control and γ2^f/f mice. Note the decreased firing frequency of FSI but not of SSC in the mutant. **c, d** Dot plots of the frequency of action potential discharges of FSI (**c**) and SSC (**d**) in control (dark red and dark blue) and γ2^f/f (light red and light blue) mice (n = 26 and n = 28 for FSI from N = 8 control mice and N = 7 γ2^f/f mice, respectively; n = 18 for SSCs from N = 3 control mice and N = 3 γ2^f/f mice; p = 0.0108 for (**c**), two-tailed Student's t test; p = 0.3987 for (**d**), two-tailed Mann–Whitney test). **e** Representative confocal images of PV⁺ interneurons (red) in control and γ2^f/f mice at P10 and P30. Scale bar: 40 μm. **f** Quantification of PV⁺ cell densities in control (dark red) and γ2^f/f mice (light red) at P10, P24, P30, and P120. Note that the increase in PV expression observed from P10 to P30 in controls is delayed and detected at P120 in γ2^f/f mice. Indeed, the number of PV⁺ interneurons is different between control and γ2^f/f mice at P24 and P30 (P10: N = 3; P24: N = 5; P30: N = 5; P120: N = 4 for controls and P10: N = 4; P24: N = 5; P30: N = 4; P120: N = 4 for γ2^f/f mice; p = 0,8421, p = 0.0230, and p = 0.0019 in control mice and p = 0.6270, p = 0.9934, p = 0.0068 in γ2^f/f mice for comparison of P10 with P24, P30, and P120, respectively; p = 0,2163 and p = 0.0176 in control mice and p = 0.1982 and p < 0.0001 in γ2^f/f mice for comparison of P24 with P30 and P120, respectively; p = 0,8781 in control mice and p = 0.0419 in γ2^f/f mice for comparison of P30 with P120; p > 0.9999, p = 0.0091, p = 0.0252, p = 0.8902 for comparisons between control and γ2^f/f mice at P10, P24, P30, and P120, respectively; two-way ANOVA test followed by a Tukey's multiple comparison post hoc test). Dot plots in (**c-f**) are presented as mean±s.e.m, dots in (**c**) and (**d**) represent data from individual FSI and SSCs, respectively, and dots in (**f**) represent data from individual mice.

increase in IPSC amplitudes from P10 to P30 in control mice, as a result of the expected increase in FSI-SSC synaptic connectivity from the second postnatal week[29], but this IPSC amplitude increase was not found in SSCs of γ2^f/f mice (Fig. 5a, b, d). Moreover, while no changes in the excitation/inhibition (E/I) ratio were observed during the second postnatal week between the two mouse lines, the E/I ratio appeared larger at the end of the first postnatal month in γ2^f/f mice, generating a strong E/I imbalance at this age (Fig. 5e, f). It is noteworthy to mention that the percentage of IPSC amplitude increase in the presence of zolpidem (1 μM), a positive modulator of γ2-GABA_ARs[6,18,30],

was equal in control and γ2^f/f mice (Supplementary Fig. 7). In contrast to the complete loss of zolpidem-sensitive currents in OPCs of γ2^f/f mice[6], this result confirmed that SSCs express functional postsynaptic γ2-GABA_ARs in both groups and that the reduced FSI-mediated inhibition arose from the early disruption of γ2-mediated synapses of OPCs. Moreover, we also found no significant differences in EPSCs, IPSCs, and E/I ratio in recorded layer IV FSI at P21-P30 in the mutant, indicating that excitation and inhibition of FSI were not altered in this group (Supplementary Fig. 8). Altogether, these results suggest that FSI myelination selectively contributes to the increase in FSI-SSC

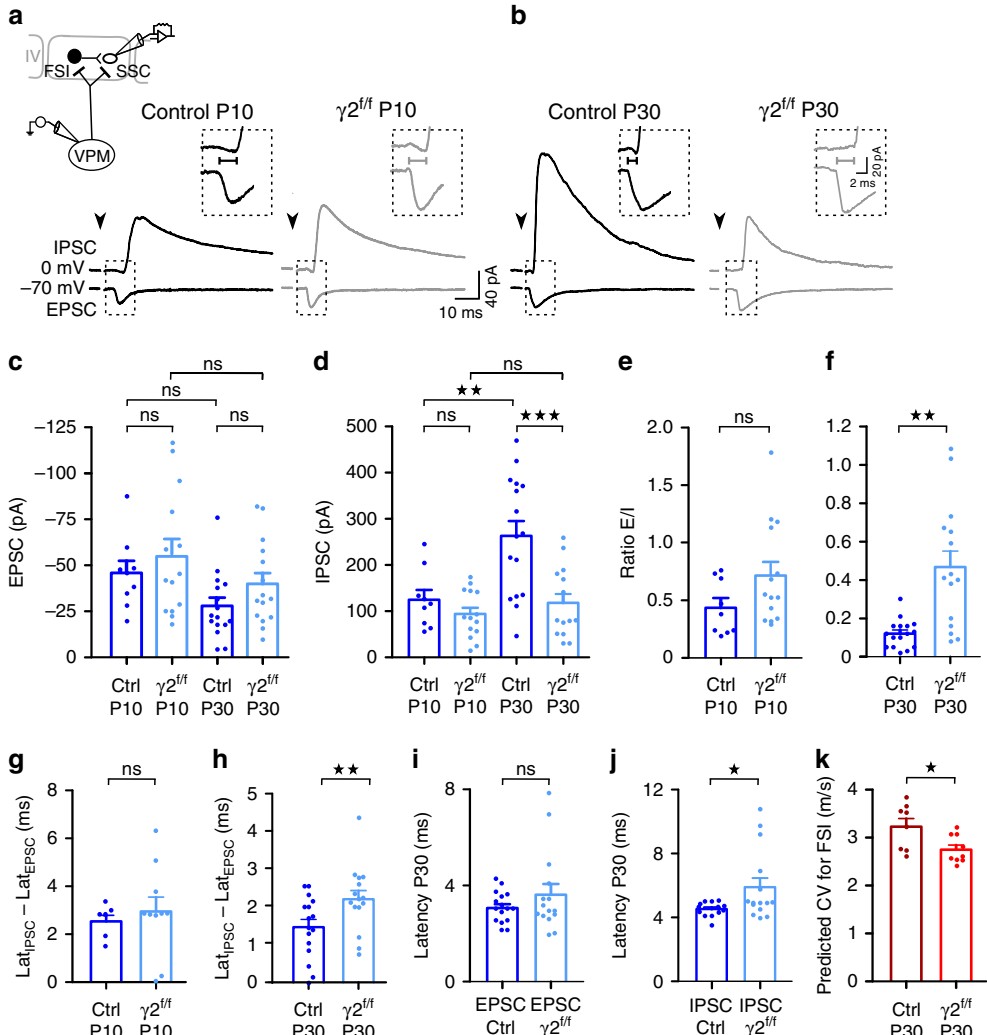

**Fig. 5 Impaired IPSC latency and strength of layer IV SSCs in γ2^f/f mice. a, b** Evoked EPSCs (bottom trace) and IPSCs (top trace) in layer IV SSCs held at −70 mV and −0 mV, respectively, in young (**a**) and mature (**b**) circuits upon thalamic stimulation in control (black) and γ2^f/f (gray) mice. Note that the delay between IPSC and EPSC latencies decreases at P30 in control but not in the mutant (lines in insets). Stimulation artifacts blanked, stimulation time indicated (arrowheads). **c, d** Dot plots of EPSCs (**c**) and IPSCs (**d**) of layer IV SSCs evoked by thalamocortical stimulation at P9-P11 (referred as P10) and P21-P30 (referred as P30) in control (dark blue) and γ2^f/f mice (light blue) (*n* = 9 and *n* = 14 at P10, *n* = 16 and *n* = 15 at P30 from *N* = 3 mice per condition; for EPSCs: *p* = 0.8443 for P10, *p* = 0.4428 for P30 between controls and γ2^f/f mice; *p* = 0.3707 for controls, *p* = 0.6511 for γ2^f/f mice between P10 and P30; for IPSCs: *p* = 0.8399 for P10, *p* = 0.0025 for P30 between controls and γ2^f/f mice; *p* = 0.0025 for controls, *p* = 0.8715 for γ2^f/f mice between P10 and P30; one-way ANOVA test followed by a Tukey's multiple comparison test). **e, f** Dot plots of E/I ratio calculated from EPSCs and IPSCs obtained at P10 and P30 for SSCs in control (dark blue) and γ2^f/f mice (light blue) (*n* = 9 and *n* = 14 at P10, *n* = 16 and *n* = 15 at P30 from *N* = 3 mice per condition; *p* = 0.1139 for (**e**), *p* = 0.001 for (**f**); two-tailed Mann–Whitney U test). **g, h** Dot plots of the delay between IPSCs and EPSCs latencies (Lat_IPSCs-Lat_EPSCs) at P10 (**g**) and P30 (**h**) in control (dark blue) and γ2^f/f mice (light blue) (*n* = 8 and *n* = 10 at P10, *n* = 16 and *n* = 15 at P30 from *N* = 3 mice per condition; *p* = 0.7396 for (**g**), *p* = 0.0076 for (**h**); two-tailed Mann–Whitney U test). **i, j** Latencies of EPSCs (**i**) and IPSCs (**j**) at P30 for the same cells in (**b**) in control (dark blue) and γ2^f/f mice (light blue) (*n* = 16 and *n* = 15 at P30 from *N* = 3 mice per condition; *p* = 0.8587 for (**I**), two-tailed Mann–Withney U test; *p* = 0.0153 for (**j**); two-tailed Student't test). **k** Predicted conduction velocity of FSI in control (dark red) and γ2^f/f mice (light red) (*n* = 8 cells from *N* = 5 control mice, *n* = 9 cells from *N* = 3 γ2^f/f mice reconstructed in Fig. 1; *p* = 0.026; two-tailed Mann–Whitney U test). Dot plots in (**c–k**) are presented as mean±s.e.m, dots from (**c–j**) represent data from individual SSCs, dots in (**k**) represent data from individual FSI.

synaptic connectivity during early postnatal development and indicate an impaired FSI-mediated inhibition of SSCs at late developmental stages.

To further explore circuit dysfunctions in the mutant, we evaluated whether the latencies of EPSC and IPSC evoked by thalamic stimulation on SSCs changed between the two mice at P10 and P30, i.e. prior to the cortical myelination process and when myelination is advanced in the barrel cortex[19,31]. Since SSCs receive a direct excitatory input and a disynaptic FSI-mediated inhibitory input during thalamic stimulation, the onset

of evoked EPSCs preceded that of evoked IPSCs at both postnatal ages (Fig. 5a, b). We found that the delay between IPSC and EPSC latencies was unchanged at P10 between the two mice while it was significantly slower at P30 in the mutant (Fig. 5a, b, g, h). In fact, while the latencies of EPSCs were similar at P30, they were significantly prolonged for IPSCs in γ2^f/f mice (Fig. 5i, j). This synaptic delay established in mature inhibitory circuits may be due to synaptic transmission defects or deficits in action potential conduction. It has been shown that synaptic latencies dependent on release mechanisms vary in an amplitude-dependent

manner[32]. In our case, however, while IPSC amplitudes largely varied in the mutant at P30, synaptic latencies were more constant (Fig. 5d, j; coefficient of variation: 0.63 and 0.37 for IPSC amplitudes and latencies, respectively). Moreover, we found that IPSC amplitudes and latencies did not correlate in this mouse line, suggesting that the delayed latency in the mutant is more likely to be due to an alteration in the conduction velocity than from deficits in neurotransmission ($n = 15$, $r = -0.3436$, $p = 0.210721$; Pearson $r$ test). To test potential FSI conduction impairments, we ran simulations of action potential propagation in these cells using the model described by Arancibia-Carcamo et al.[21], which predicts the conduction velocity of action potentials according to the size of nodes and internodes (see Online Methods). To run this computational model, most morphological and electrophysiological parameters describing FSI axons were set to fit those recently described for myelinated PV$^+$ interneurons[7,9] (Supplementary Table 2). The resting potentials, node and internode lengths were set to the values obtained experimentally from each reconstructed FSI in control and γ2$^{f/f}$ mice (Fig. 1). Based on these simulations, we found that FSI axons of γ2$^{f/f}$ mice had slower predicted conduction velocities than those of control mice (Fig. 5k). Although we cannot totally rule out that neurotransmission deficits of presynaptic FSI contributed to the latency delay of IPSC responses in the mutant, the experimental latency retardation -that is independent of IPSC amplitudes- and the slow conduction velocity obtained by computational simulations support the fact that the abnormal longer nodes and internodes of FSI affect their ability to rapidly conduct action potentials.

Finally, the axonal arborization of most layer IV FSI is largely confined within a single barrel structure with some collaterals projecting to layers II/III, V and the adjacent barrel[33]. Therefore, the myelinated axon region of layer IV FSI, which is biased towards the proximal part[9], mainly lies within the barrel where FSI displays a very high connection probability with SSCs[34]. To test if the decrease in IPSCs evoked by thalamic stimulation at P30 was induced by a deficit on the maturation of the connectivity between nearby FSI and SSCs within single barrels, we compared the FSI-SSC connection probability using paired recordings between cells displaying inter-somatic distances not exceeding 40 μm (Fig. 6). We found that the stimulation of presynaptic FSI elicited IPSCs on neighbor SSCs in 80% of tested pairs in control mice while none of the tested pairs were connected in γ2$^{f/f}$ mice (Fig. 6a–c). Hence, a defect in the inhibitory connectivity of SSCs located in the vicinity of the myelinated region of FSI axons accounts for the significant decrease in the feedforward inhibition of SSCs.

Overall, our results show that myelination defects of FSI are associated with a reduction in the firing frequency and conduction velocity of these interneurons as well as to a deficient FSI-SSCs synaptic connectivity, affecting feedforward inhibition without perturbing excitatory thalamocortical circuits. Since the robustness and high temporal precision of the FSI-mediated feedforward inhibition onto SSCs is necessary to avoid a redundant recurrent excitation and maintain a dynamic balance between excitation and inhibition[12,35], our results point to a central role of FSI myelination in regulating the functioning of mature cortical sensory inhibitory circuits.

**Whisker-based texture discrimination deficits.** Layer IV FSI provides a powerful feedforward inhibition to SSCs which sparsely but reliably respond to sensory inputs. Such information coding is essential for sensory perception during whisker-related behaviors[12]. To determine whether FSI myelination defects and

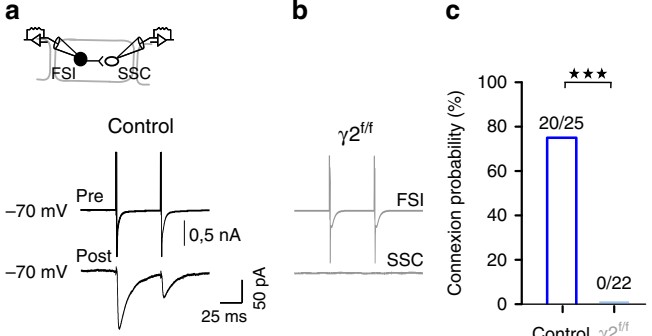

**Fig. 6 Impaired synaptic connectivity between FSI and SSC in layer IV of γ2$^{f/f}$ mice. a, b** Paired recordings between a presynaptic FSI (top trace) and a postsynaptic SSC (bottom trace) in layer IV of the barrel cortex in control (**a**) and γ2$^{f/f}$ mice (**b**). **c** Connection probabilities for paired recordings between FSI and SSCs in control (dark blue) and γ2$^{f/f}$ mice (light blue) ($n = 25$ pairs from $N = 6$ control mice and $N = 22$ pairs from $N = 5$ γ2$^{f/f}$ mice; $P < 0,0001$, two-sided Chi-squared test). Note the lack of connectivity between FSI and SSC in γ2$^{f/f}$ mice.

cortical sensory inhibitory circuit dysfunctions are associated with behavioral impairments during sensory experience, we subjected adult mice to a whisker-dependent texture discrimination task[36]. Based on the ability of mice to discriminate an object when their whiskers come in contact with it, this behavioral test comprised three different phases: habituation to promote exploratory behavior, learning through the exploration of two identical textured objects with vibrissae and testing of the discriminative whisking of a new textured object (Fig. 7a). For analysis, we considered only mice which explored the two objects for more than 2 s during the learning and testing phases, as previously described[36]. We found that 4 out of 14 control mice and 11 out of 19 γ2$^{f/f}$ mice lacked sufficient exploration of the textured objects and were thus excluded. To test whether this insufficient object exploration was due to a lower overall exploratory behavior in the arena, we tested the exploratory behavior in a conventional open-field. We found no significant differences between control and γ2$^{f/f}$ mice on the distance and time spent in the outer and inner zones, the total time of activity and inactivity and the time course of traveled distance (Supplementary Fig. 9). The lack of object exploration was therefore most likely due to a lack of interest for the objects rather than a deficient exploratory capacity. In addition, we did not observe significant differences in whisker-based texture exploration during the learning phase between control and γ2$^{f/f}$ mice, which ensures comparable levels of exploratory behavior among mice included in the analysis (Fig. 7b). We then checked the ability of these mice to discriminate between two different textures during the testing phase (Fig. 7a). We observed that control mice were indeed able to discriminate between the two textures, as they spent significantly more time exploring the novel textured object compared to the textured object they had already encountered (Fig. 7c). On the contrary, γ2$^{f/f}$ mice could not discriminate between the two textures as they spent similar amounts of time exploring the novel and old textured objects (Fig. 7c). In fact, the time spent investigating the novel texture was significantly lower in the mutant compared to controls (Fig. 7d). It is worthy to mention that mice from both groups considered for analysis spent similar amounts of time exploring the textures during the testing phase, confirming that differences in novel texture exploration were due to the ability to discriminate and not a by-product of the amount of time spent exploring the objects (exploration time: 9.74 ± 1.66 s for control mice, 9.06 ± 2.33 s for γ2$^{f/f}$ mice; p = 0.809, two-sample Student's

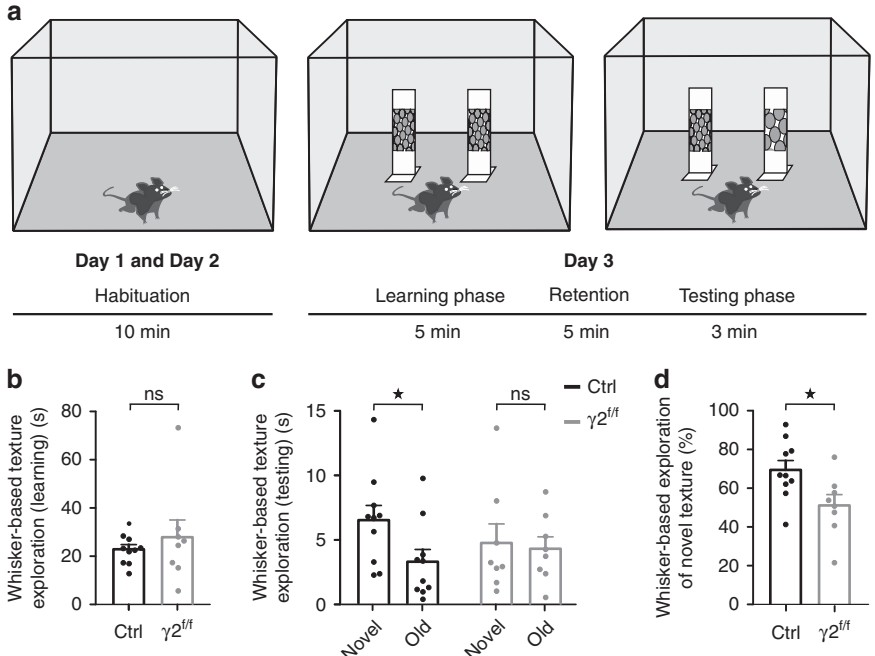

**Fig. 7 Impaired whisker-based texture discrimination in γ2$^{f/f}$ mice. a** Schematic of the whisker-dependent discrimination task. **b**, **c** Dot plots of the whisker-based texture exploration time during the learning (**b**) and testing (**c**) phases in control (black) and γ2$^{f/f}$ mice (gray) ($N = 10$ and $N = 8$ for control and γ2$^{f/f}$ mice, respectively; $p = 0.696$ for (**b**), two-tailed Mann–Whitney U test; $p = 0.033$ and $p = 0.742$ for (**c**) for control and γ2$^{f/f}$ mice, respectively; two-tailed paired Student's $t$ test and two-tailed Mann–Whitney U test, respectively). Note that control mice preferentially explore the novel object while mutants do not discriminate between the two objects. **d** Dot plots of the percentage of whisker-based exploration of a novel texture in control (black) and γ2$^{f/f}$ mice (gray) ($N = 10$ and $N = 8$ for control and γ2$^{f/f}$ mice, respectively; $p = 0.0235$, two-tailed paired Student's $t$ test). Dot plots are presented as mean±s.e.m and dots represent data from individual mice.

t-test). These behavioral data thus indicate that a disruption of FSI-oligodendroglia interactions, inducing severe FSI myelination defects during development, also leads to deficient whisker-dependent discrimination of a new texture in the adult.

## Discussion
For a long time, it has been known that myelination is important to speed up the conduction of action potentials. However, myelination patterns vary according to CNS regions and it is unknown why some axons are ensheathed by myelin while others are not. Our current understanding of how myelin contributes to the fine-tuning of neuronal networks in the CNS is very limited. Particularly, an increasing number of studies has revealed an unexpectedly extensive myelination of short-range PV$^+$ FSI[6–10], but the relevance of FSI myelination in local circuits remains unclear. This report focuses on the impact of FSI myelination on the morphology and function of FSI in a specific inhibitory circuit of layer IV of the barrel cortex, which is critical for the sensory gating process[12]. Our results demonstrate that, beyond conduction, FSI myelination shapes the morphology of the proximal axon, the high frequency of action potential discharges and the connectivity of these interneurons with excitatory neurons. These deleterious effects on FSI are then associated with impairments in whisker-dependent discrimination, i.e. the outcome of sensory neuronal processing.

In layer IV of the somatosensory cortex, virtually all myelinated GABAergic interneurons are FSI[9]. These interneurons exhibit a myelin distribution biased towards the proximal axon and display a single internode before the first axonal branch point appearing around 50 μm[9]. They also display shorter internodes and nodes than non-GABAergic neurons[7,9]. Here, we found that an early disruption of FSI-oligodendroglia interactions results in a

defective myelination, which displayed almost twice longer nodes and internodes and were associated with aberrant modifications of proximal axon morphology. Hence these results point to a critical role of FSI myelination in guiding proximal axon morphology. Conversely, a recent report shows that FSI morphology is important to guide myelination[37]. The authors show that the enlargement of FSI size by genetic manipulations increases axonal myelination, while a decreased cell size leads to a reduced myelination[37]. Therefore, it exists an interdependency between FSI morphology and myelination that reveals a previously unexpected and intimate interplay between FSI and oligodendroglia which is critical for maintaining both the architecture of FSI axons and a correct myelination pattern. This interdependency probably already exists early on during postnatal development as FSIs synaptically contact OPCs during the first two postnatal weeks[8,14]. Although γ2-mediated GABAergic synapses of OPCs do not impact the proliferation and differentiation of these progenitors at P10 (Supplementary Fig. 1 and Ref. [6]), the early inactivation of the γ2 subunit of GABA$_A$ receptors in these progenitors induces long-term adverse effects on oligodendroglia by reducing OPC density[6]. Interestingly, the reduced morphological complexity of differentiating OLs in the mutant also suggests that the inactivation of OPC GABAergic synapses mainly affects the transition from a progenitor to an OL state. In turn, we observed an aberrant axon morphology and myelination of FSI. The idea that interneurons and oligodendroglia are reciprocal partners during development is reinforced by recent data showing that (1) both cell types are born from progenitors expressing similar transcription factors and lying in the same germinal regions[8,38,39]; (2) lineage-related interneurons and OPCs are initially over-produced and then significantly demised at early postnatal stages[8,38,40]; (3) surviving lineage-related interneurons

and oligodendroglia form anatomical and functional clusters at postnatal stages[8], and (4) migrating interneurons secrete the cytokine fraktaline which promotes oligodendrogenesis via the fraktaline receptor CX3CR1 expressed in OPCs[41].Interestingly, different axon geometries and myelin distributions at proximal axon regions also shape action potential discharges allowing intrinsic excitability of neurons to be tuned[20,23,42]. Compared to pyramidal neurons whose action potentials are initiated far from the axon hillock[43], the generation of action potentials in FSI occurs very close to the soma[13]. Despite the small axon diameter and extensive branching of FSI axons, which should limit spike generation and propagation, the unique fast-spiking phenotype of these interneurons is known to be ensured by the selective expression of ion channels such as Kv3-type K$^+$ channels[44] and a supercritical axonal density of Na$^+$ channels[13]. However, it is probably not a coincidence that FSI myelination occurs in the proximal part of the axon where action potentials are initiated and efficiently propagated towards smaller axon branches and synaptic termini[7,9]. In fact, myelination defects of FSI caused a significant reduction in their high firing frequency without modifying other properties such as input resistance, resting potential, AHP, action potential amplitudes and fast spike kinetics. This lack of changes in intrinsic electrophysiological properties makes major modifications in protein and channel expression at the AIS and nodes of Ranvier in the mutant rather unlikely. Future studies, however, will be needed to determine the molecular identity and distribution of different proteins and ion channels in myelinated FSI axons both in normal and pathological conditions. Nevertheless, although an increase in internode length tends to increase action potential conduction[45], aberrant longer internodes and nodes like those observed in γ2$^{f/f}$ mice result in a decreased predicted conduction velocity (Fig. 5k). The slow conduction could thus be the main cause of the reduced firing frequency of FSI in the mutant. Although there is a relationship between the lengths of internodes and nodes and conduction speed, node lengths appear to play a critical role in tuning action potential propagation[21] which may explain the reduced conduction velocity of FSI in the mutant. Surprisingly, a large number of axon branch points also appeared myelinated in γ2$^{f/f}$ mice, a severe defect which must also compromise the proper propagation of action potentials[20]. In summary, the specific proximal axon morphology and myelination pattern of FSI mainly impacts the high rate and reliable propagation of action potentials rather than the spike form. Although myelin distribution is restricted to the first part of FSI axons, our results show that this myelination pattern is required to optimize the fast-spiking phenotype and the high temporal precision of action potential discharges of FSI. Therefore, these properties do not rely solely on FSI intrinsic properties. Finally, although we cannot totally rule out that myelination defects occur in other neurons in the cortex of γ2$^{f/f}$ mice, we did not observe changes in the electrophysiological properties of SSCs and pyramidal neurons as well as in the amplitudes and latencies of EPSCs evoked in SSCs by direct thalamic inputs (while those of FSI-mediated IPSCs were reduced). Thus, if myelination profiles are also altered in other neurons in the mutant, these changes do not appear to significantly interfere with their physiological properties.

Recently, a study demonstrated that a lack of compact myelin correlates with a reduced GABAergic synaptic transmission in Purkinje cells of the cerebellum[46]. Diminished cerebellar myelination following neonatal hypoxia also decreases GABAergic synaptic activity of migrating white matter interneurons[47]. These reports suggest that, beyond conduction, axonal myelination influences synaptic neurotransmission. In line with this, we found that the inactivation of FSI–OPC synapses during early postnatal development resulted in myelination defects of FSI that were associated with a strong reduction of FSI-SSC connectivity in mature local circuits confined within the barrel structure. While the feedforward inhibition of SSCs was significantly reduced in the mutant upon electrical thalamic stimulation at P30, our paired recordings revealed a lack of FSI-SSC connectivity within a single barrel and in the area of the myelinated part of the FSI axon (i.e. <40 μm intersomatic distances). Distal FSI (e.g. FSI in adjacent barrels or layers) as well as some somatostatin-expressing interneurons[48] may thus participate in the remaining IPSCs evoked by electrical thalamic stimulation at P30 in the mutant. On the contrary, the excitatory thalamocortical input onto SSCs remained unperturbed, showing that OPC synapse inactivation resulted in a specific decrease of FSI-mediated inhibition, causing a strong excitation/inhibition imbalance. Interestingly, while the inhibition of SSCs was similar between control and γ2$^{f/f}$ mice prior to myelination onset (i.e. P10), this connectivity in the mutant did not follow the developmental increase normally occurring in conjunction with myelination[19,29]. Since GABA switches from a depolarizing to a hyperpolarizing neurotransmitter around P8 in the cerebral cortex[49] and reduced IPSCs occurred in SSCs but not FSI at a later developmental stage in γ2$^{f/f}$ mice, it is very likely that a decreased FSI-mediated inhibition exists in SSCs in vivo in the mutant. Hence, our results point to a crucial role of FSI myelination in the maturation of synaptic connectivity of these interneurons with nearby SSCs within barrels.

In the barrel cortex, the feedforward inhibition of FSI in layer IV circuits is key to avoid redundant recurrent thalamocortical excitation of SSCs and maintain the dynamic balance between excitation and inhibition necessary to optimize sensory responses[12]. Indeed, the synchronous and fast discharge of FSI, immediately following thalamocortical excitation, generates a feedforward release of GABA within layer IV barrel structures that shortens the duration of SSC excitation[35]. By suppressing most of the excitability of SSCs, this feedforward inhibitory circuit filters irrelevant information and optimizes sensory processing, allowing for a reliable and precise response to relevant stimuli[12,35]. Our findings show that FSI dysfunctions caused by myelination defects impact the functioning of these interneurons in a cortical neuronal circuit that is critical for sensory information processing during whisker-related behaviors. Notably, FSI constitutes a primary source of inhibition in the neocortex and provide a robust synaptic input which inhibits the activity of excitatory neurons with high temporal precision, shaping cortical circuit dynamics during specific brain states and different behavioral contexts. Beyond conduction, our data demonstrate that FSI myelination appears as a key factor in regulating the inhibition of local cortical networks. Considering that the total axon length of layer IV FSI in the rodent somatosensory cortex is comprised from 10 mm to 22 mm[33] and that FSI myelination is distributed within ~0.4 mm of the proximal axon (Supplementary Fig. 2d and ref. [9]), only 2.5-4% of the total axon length of these interneurons is myelinated. Nevertheless, myelin anomalies of these interneurons cause dramatic changes in firing frequency, inhibitory circuit function, and behavior.

Interestingly, FSI constitutes a recurrent locus of dysfunctions in neurodevelopmental diseases such as schizophrenia. Notably, the synchronization of neuronal ensembles in the gamma range frequency, which largely depends on FSI activity[50], is commonly altered in this disease. Moreover, a recent study performed in a rat model of schizophrenia showed a hypomyelination of cortical PV$^+$ interneurons[51]. Considering our demonstration of the important role of FSI myelination in regulating the function of FSI, it is thus possible that FSI myelination defects alter local cortical circuit oscillations in vivo which, in turn, contribute to cognitive deficits observed in this disease[5,15].

## Methods

**Transgenic mice and induction of Cre-mediated recombination**. All experiments followed the European Union and institutional guidelines for the care and use of laboratory animals and were approved by the French ethical committee for animal care of university Paris Descartes (Committee N°CEEA34) and the Ministry of National Education and Research (Project No: 13094-2017081712355709). For experiments, we crossed transgenic mouse lines for NG2creERT2, Gcamp3[LoxP/LoxP], and $\gamma2^{LoxP/LoxP}$ as described[6]. NG2creERT2[+/-];Gcamp3[f/f];$\gamma2^{f/f}$ mice ($\gamma2^{f/f}$ mice) were used to inactivate $\gamma2$-GABA$_A$R-mediated synapses in OPCs while the NG2creER[+/-];Gcamp3[f/f] mice were used as controls[6]. Animals were genotyped by PCR using primers specific for the different alleles and Cre expression was induced from P3 to P5 by daily intraperitoneal injections of 0.2 mg of 4-OHT (Tocris Bioscience) as previously described[6]. Animals were maintained in the animal facility under 12 hours light/dark cycle with ad libitum access to food and water and in a controlled average ambient temperature of 21 °C and 45% humidity. Both female and male were indiscriminately used at different postnatal stages from P10 to P120.

**Acute slice preparation and electrophysiology**. Electrophysiological experiments were performed in the barrel cortex using either acute parasagittal slices of 300 μm-thick or acute tangential thalamocortical slices of 350-μm thick[14]. Recorded cells were visualized with an iXon1 14-bit digital camera (Andor Technology, UK) and the Imaging Workbench version 6.0 software (Indec Biosystems, USA) under an Olympus BX51 microscope equipped with a 40X fluorescent water-immersion objective. Excitation light to detect fluorescent OL lineage cells was provided by Optoled Light Sources (Blue Optoled; Cairn Research, UK). Excitation and emission wavelengths were obtained by using 470 and 525 nm filters, respectively.

Patch-clamp recordings were performed at RT using an extracellular solution containing (in mM): 126 NaCl, 2.5 KCl, 1.25 NaH$_2$PO$_4$, 26 NaHCO$_3$, 20 glucose, 5 pyruvate, 3 CaCl$_2$ and 1 MgCl$_2$ (95% O$_2$, 5% CO$_2$). Biocytin-loaded FSI for morphological analysis and presynaptic FSI during paired recordings were recorded with an intracellular solution containing (in mM): 130 Kgluconate (KGlu), 10 GABA, 0.1 EGTA, 0.5 CaCl2, 2 MgCl2, 10 HEPES, 2 Na2-ATP, 0.2 Na-GTP, 10 Na2-phosphocreatine and 5 biocytin (pH ≈ 7.3). Biocytin-loaded OPCs and differentiating OLs for morphological analysis and postsynaptic SSCs during paired recordings were recorded with a CsCl-based intracellular solution containing (in mM): 130 CsCl, 5 4-aminopyridine, 10 tetraethylammonium chloride, 0.2 EGTA, 0.5 CaCl[2], 2 MgCl2, 10 HEPES, 2 Na2-ATP, 0.2 Na-GTP, 10 Na$_2$-phosphocreatine and 5 biocytin (pH ≈ 7.3). Thalamocortical feedforward excitation and inhibition were recorded in layer IV OPCs, SSCs, and FSI at different postnatal ages. Excitatory and inhibitory postsynaptic currents were evoked by a bipolar concentric electrode placed in the thalamic nucleus at −70 mV and 0 mV, respectively (100 ms pulse, 5–40 V; Iso-Stim 01D, npi electronic GmbH, Tamm, Germany), with a CsMeS-based intracellular solution containing (in mM): 125 CsCH$_3$SO$_3$H, 5.4-aminopyridine, 10 tetraethylammonium chloride, 0.2 EGTA, 0.5 CaCl[2], 2 MgCl$_2$, 10 HEPES, 2 Na$_2$-ATP, 0.2 Na-GTP and 10 Na$_2$-phosphocreatine (pH ≈ 7.3). Recordings were made without series resistance (R$_s$) compensation. R$_s$ was monitored during recordings and cells showing a change of more than 30% were discarded. Potentials were corrected for a junction potential of −10 mV when using KGlu- and CsMeS-based intracellular solution. Acquisition was obtained using Multiclamp 700B and pClamp10.2 software, filtered at 4 kHz and digitized at 20 kHz. Firing properties, evoked postsynaptic currents and paired recordings were analyzed off-line using pClamp10.5 software (Molecular Devices), and Neuromatic package[52] within IGOR Pro 6.0 environment (Wavemetrics, USA) as previously described[8,14].

**Immunostainings and cell countings**. Mice were perfused with phosphate buffer saline (PBS) followed by 0.15 M phosphate buffer (PB; pH ≈ 7.3), containing 4% paraformaldehyde (PFA) at P10, P24, and P30 (n = 3 to 5 mice per age and condition). Brains were kept 2 h in PFA and stored in PBS at 4 °C before cutting. Next, coronal vibratome slices (100 μm) were cut in PBS ice-cold solution (4 °C), permeabilized with 0.2% triton X-100 and 4% Normal Goat Serum (NGS) for 1 h and incubated with primary antibodies diluted in a 0.2% triton X-100 solution and 5% NGS one night at P10 and three nights for other ages. Primary antibodies used for immunohistochemistry were rabbit polyclonal anti-Olig2 (1:400; ref. AB9610, Millipore), chicken polyclonal anti-GFP (for detection of Gcamp3 here used as a reporter; 1:1000; ref. A10262, ThermoFisher Scientific), rabbit polyclonal anti-PV (1:1000; ref. PV-27, Swant), mouse monoclonal anti-APC (CC1; 1:100; clone CC-1; ref. OP80, Millipore), rabbit polyclonal anti-NG2 (1:100; ref. AB5320, Thermofisher Scientific), rat monoclonal anti-MBP (1:100; clone 26; ref. AB7349, Abcam), mouse monoclonal anti-NeuN (1:250; clone A60; ref. MAB377, Merck) and mouse monoclonal anti-Ankyrin G (1:100; clone N106/36; ref. MAB1683, Merck). Slices with primary antibodies were washed three times in PBS and incubated in secondary polyclonal antibodies. We used goat anti-rabbit DyLight-405 (ref. 35551; A-11039, Thermofisher Scientific), goat anti-chicken Alexa Fluor-488 (ref. A-11039, Thermofisher Scientific), goat anti-mouse Alexa Fluor-546 (ref. A-11030, Thermofisher Scientific), goat anti-mouse DyLight-633 (ref. GTX76787; Genetex) or goat anti-rat Alexa Fluor-633 (ref. A-21094, Thermofisher Scientific) at RT for 2 h at 1:500 or 2 days at 1:200 according to the age of the animal.

Optical sections of confocal images were acquired using 63X oil objectives with a SP8 Leica confocal microscope and the LAS X software (version 3.7.2.22383). Images were processed and analyzed using NIH ImageJ software (version 1.52i)[53]. Cell countings were performed manually in cortical layer IV with the ROI manager tool of ImageJ. We analyzed four slices per animal using 350 ×250 μm of 20 z-sections (each 1 μm). To prevent border effects in countings, cells that were at the boundaries of the analyzed volume were not considered in three of the six sides of the cube if somata were not fully inside. Cell density was obtained by dividing the number of cells by the cube volume.

**Confocal 3D reconstructions**. Immunostainings of slices containing biocytin-loaded cells were permeabilized with 1% triton X-100 and 10% Normal Goat Serum (NGS) overnight and incubated four nights with primary antibodies diluted in a 1% triton X-100 solution and 10% NGS. Primary antibodies used for immunohistochemistry were either rabbit polyclonal anti-PV (1:1000; ref. PV-27, Swant) or rabbit polyclonal anti-Caspr (1:500; ref. ab34151, Abcam) and rat monoclonal anti-MBP (1:100; clone 26; ref. AB7349, Abcam). Slices with primary antibodies were washed three times in PBS and incubated in secondary antibodies coupled, respectively, to goat anti-rabbit Alexa Fluor-405 and goat anti-rat Alexa Fluor-633 (1:200) as well as in conjugated streptavidin–Alexa Fluor-546 (1:200; ref. S11225) at 4 °C for 2 days (ThermoFisher scientific).

Axonal reconstructions of biocytin-loaded FSI at P25-P30 were obtained from confocal images using a 63X oil objectives with a SP8 Leica confocal microscope. Biocytin-loaded FSI was randomly selected during recordings. To avoid any biased during analysis, a successful axon labeling was the only criterion to keep the cell for further morphological analysis. FSI axon segments were considered as myelinated if they co-expressed MBP (125 z-sections; each 0.3 μm). Semi-automatic 3D reconstructions of FSI myelinated axons were obtained using Simple Neurite Tracer plugin in NIH ImageJ software[53] which allowed us to quantify the total axon length, the number of axon branches, the number, distribution and length of MBP$^+$ internodes. FSI axon morphology was determined from 3D reconstructions where only clearly interconnected branches were considered. Although this choice left aside the most distal and thinnest processes, it ensured that no small dendritic branch was included by error in the two groups. This procedure also reduces possible variations due to slicing, cell loading, and depth of recorded cells in slices, ensuring a more accurate comparison between groups. For node length analysis, we selected nodes found between two consecutive myelinated internodes flanked by two Caspr-labeled paranodes. A maximum intensity projection was generated from sections including Caspr labeling for a single node (up to four interleaved confocal slices of 0.30 μm intervals), and a line intensity profile was generated across both Caspr-labeled paranodes (Fig. 2c). The size of the node was then calculated measuring the distance between the half maximum intensity for each paranode[21]. Finally, OPCs and pre-myelinating OLs were identified by their fluorescence and their different I-V curves in acute slices at P30-P32[22]. To characterize the complexity of the ramifications of layer IV OL lineage cells in relation to the distance from the soma, we performed Sholl analysis with ImageJ software using confocal z-stack images of biocytin-loaded cells (125 optical sections of 0.3 μm each).

**Conduction velocity simulations**. To simulate action potential propagation of reconstructed biocytin-loaded FSI in control and $\gamma2^{f/f}$ mice, we ran custom-made scripts on Matlab (version: R2018b; Mathworks, Natick, MA), made freely available from Arancibia-Carcamo et al. (2017)[21]. Briefly, the model was implemented from Model C of Richardson et al.[54]. It contains an axon divided into three compartments (i.e., node, paranode and internode) across which current flows through voltage-gated ion channels present in the membrane of the different compartments. Parameters describing the morphology of the axon were set to fit those recently described for myelinated PV$^+$ interneurons (Supplementary Table 2)[7,9]. The node and internode lengths were set to values obtained experimentally for control and $\gamma2^{f/f}$ mice (Supplementary Table 3). We assumed that the number of sodium channels in a single node remained the same for the two groups. Electrophysiological parameters were the same as in Arancibia-Carcamo et al.[21], except for resting potentials which were experimentally recorded (Supplementary Table 3).

**Behavioral assays**. The whisker-dependent texture discrimination task, specifically designed to assess somatosensory function[36], was divided into three phases: habituation, learning, and testing (Fig. 7a). The habituation phase consisted of 2 sessions of 10 minutes spread over two days during which adult mice were free to explore a 40 cm × 40 cm arena covered with 2 cm of cage bedding. The learning and testing phases occurred on the third day. During the learning phase, two identical rectangular objects of plexiglass (4 cm × 15 cm) containing sandpaper on both sides (180 grit; 7.5 cm height) were introduced at equal distances from the walls of the arena. Mice were first free to explore the objects for 5 min and then placed alone into a transport cage for other 5 min (i.e., retention) in order to control for a low involvement of the hippocampus[36]. Finally, during the testing phase, the two objects were removed and replaced with one identical to the first two and one with a slightly thicker sandpaper (120 grit). Mice were then put back into the testing arena and left to explore the two objects for 3 min. Mice were

considered to be exploring an object when they were in its close vicinity, nose directed towards or touching the object. Resting, digging, and grooming around the object was not considered as whisker-based exploration. Mice that did not explore one object or explored for less than 2 seconds either during the learning or testing phase were excluded from the analysis for lack of appropriate exploratory activity.

For the open field assay, mice were placed for 30 minutes in a 50 cm ×50 cm arena. The movements of the animals were tracked using an infrared camera and automatically processed with the ViewPoint behavioral analysis software (vesrion 2011, ViewPoint). Distance traveled, velocity, and time spent in predefined zones were established.

**Statistics**. Data are expressed as mean ± SEM from n cells, paired recordings or animals. GraphPad Prism (version 5 and 8; GraphPad Software Inc., USA) were used for statistical analysis. Each group of data was first subjected to D'Agostino–Pearson normality test. According to the data structure, two-group comparisons were performed using a two-tailed Student's $t$ test or a Mann–Whitney U test for unpaired samples and a Wilcoxon signed-rank test for paired samples. Multiple group comparisons were performed using the parametric one-way ANOVA followed by post-hoc Tukey's test or the non-parametric Kruskal–Wallis test followed by a post-hoc Bonferroni's test. For small statistical groups (less than 5), we assumed normality and systematically performed parametric tests. Only an exceptional behavioral data point submitted to Grubbs test was excluded as an extreme outlier. Connection probabilities between control and $\gamma 2^{f/f}$ mice were tested with a Chi-squared test. Correlation between IPSC amplitudes and latencies were tested with a Pearson r test and differences were considered significant when $P < 0.05$.

**Reporting summary**. Further information on research design is available in the Nature Research Reporting Summary linked to this article.

## Data availability

Source data are provided with this paper.

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

## Acknowledgements
The authors thank Fernando C. Ortiz for critical reading of the manuscript. We also thank the imaging NeurImag and behavior Phenobrain facilities of IPNP and their funding sources (Fédération pour la Recherche Médicale, Fondation Leducq). This work was supported by grants from Fondation pour la Recherche Médicale (FRM, Equipe FRM DEQ20150331681), Agence Nationale de la Recherche (ANR, ANR-14-CE13-0023), Fondation pour l'aide à la recherche sur la Sclérose en Plaques (ARSEP), ANR under the frame of Neuron Cofund (Era-Net Neuron, project No. R19068KK) and a subaward agreement from the University of Connecticut with funds provided by Grant No. RG-1612-26501 from National Multiple Sclerosis Society (NMSS). M.V. is recipient of a PhD fellowship from Région Île-de-France and is supported by the FIRE doctoral school-Programme Bettencourt. M.B. was recipient of PhD fellowships from Université Paris Descartes and FRM. M.C.A. is a CNRS (Centre National de la Recherche Scientifique) investigator.

## Author contributions
N.B. conducted electrophysiological experiments, immunostainings, and analysis of data. M.V. performed behavioral experiments, immunostainings, and conduction velocity simulations and M.B. contributed with immunostainings. N.B., M.V., and M.C.A. designed experiments. All authors wrote the manuscript. M.C.A. supervised the project.

## Competing interests
The authors declare no competing interests.
