## [Peer Review File · Nature Communications]

Reviewers' comments:

Reviewer #1 (Remarks to the Author):

In the manuscript "Myelination of parvalbumin interneurons shapes the function of cortical sensory inhibitory circuits", Benamer et al describe exciting new findings regarding the role of GABAergic neuron-glia signaling for myelination of axons of cortical interneurons. The authors report that reduction of GABAergic signaling between interneurons and oligodendrocyte progenitor cells, OPCs, (via deletion of gamma-2 subunit of GABAARs in OPCs) results in altered length of myelinated internodes and nodes of Ranvier in interneurons, changed firing pattern of interneurons with affected axonal myelination, and deficits in whisker-based texture discrimination. At present, mechanisms and functional significance of myelination of cortical interneurons during health and diseases remain poorly investigated. The study of Benamer et al sheds new light on the topic and inspires further research. I think that the study is well designed and carefully performed. It employs a combination of various approaches, from single-cell recordings to testing behavior of animals, and this design makes the conclusions solid. Yet I have few questions:

(1)The authors report that axons of interneurons in gamma-2fl/fl mice have longer nodes and internodes. Do the authors know whether the length of the axon of interneurons is altered in these mice? If the axonal length remains un-changed, longer nodes/ internodes probably indicate that fewer OLs are involved in myelination of each axon. And this in turn may mean that less OLs are generated (and are available for myelination) in gamma-2fl/fl mice, and/or that OLs in these mice are less eager to myelinate (because of e.g. energetic failure or other defects). Is it possible to reliably measure/estimate the length of the axons with altered myelination in gamma-2fl/fl mice, and also to count/estimate the number of OLs available to myelinate these axons? Estimating the length of the axon would also be interesting because the authors propose that FSI in gamma-2fl/fl mice may be less mature (page 9). May be maturation and axonal length are inter-related somehow?

(2)When the authors report the data regarding myelination defects in gamma-2fl/fl mice, they show that n = 8 cells (Figure 1) which may appear as a bit too low number. Do those cells represent typical examples of how FSI look like in these mice or do they rather represent a sub-population? How easy/difficult is it to find the interneurons with altered myelination in gamma-2fl/fl mice? Are there also FSI with normal myelination in these animals? How easy is it to find those? How the number of FSI with altered myelination compares to the number of FSI with unchanged myelination? In addition, it is also not clear from how many animals the eight cells have been acquired? It is important to provide all these pieces of information in order to clarify for the reader whether the authors describe typical changes in the whole population of FSI, or whether they rather show the sub-population of cells in which myelination defects have occurred.

(3)Please, also indicate the number of animals AND the number of cells in all Figures (or Figure Legends).

(4)The authors report that differentiated OLs in gamma-2fl/fl mice have altered morphology which they present in Figure 3. But are those cells really OLs? Why don't they have myelin sheathes?

(5)The authors present the E/I ratio and some alterations in it. These experiments are interesting, but they are performed in whole-cell mode with "artificial" chloride concentration. Hence, it is not

clear how these findings reflect the real situation regarding the balance between excitation and inhibition in vivo, where chloride concentration in interneurons may be different from the one in the recording pipette, and it may also depend on the age, and it may also be altered in gamma-2fl/fl mice (it is clear that only GABARs in OPCs are expected to be affected, but a feed-back mechanism from myelinating cells to neurons may exist). Please, explain/comment on this issue.

(6) When the authors test for relationship between altered firing of FSI and myelination impairments, they analyze electrophysiological properties of spiny stellate cells for comparison. But those cells are excitatory neurons (to my knowledge). What is the rationale for using SSCs for comparison in the described experiments? Would not it be more logical to compare the firing pattern of FSIs with affected myelination to the firing pattern of other cortical interneurons whose myelination is (presumably) not affected in gamma-2fl/fl mice?

(7) The "Discussion" part is presented largely as expanded repetition of the "Results" section. I think it could be shortened. And it would be interesting to include some thoughts or inspiring ideas for future research. For instance: What could be the mechanisms through which axon-OPC signaling regulates/modulates firing pattern of interneurons? Is it a kind of feed-back mechanism? Signaling from axons to the soma? Can it be that longer nodes of Ranvier "attract" more Na-channels to the axon, and this in turn results in diminished number of Na-channels in the neuronal soma and in the axon-hillock, and hence in altered firing? What is already known regarding alterations of myelination of interneurons during diseases? How those alterations compare to the present findings of the authors?

Reviewer #2 (Remarks to the Author):

Benamer et al. present an intriguing study that provides novel insights into the roles of GABAergic OPC synapses and myelination in cortical circuit development. They combine electrophysiological, post-hoc immunohistochemical, and behavioral methods to establish distinct morphological and behavioral phenotypes associated with oligodendrocyte lineage deletion of the gamma-2 subunit of GABAARs. Furthermore, this manuscript aims to establish a bidirectional signaling mechanism, through which fast-spiking interneurons (FSI) -> OPC synaptic input regulates not only the morphology of oligodendrocytes and myelination of FSI, but also the axonal morphology and number of PV-positive interneurons in the cortex. The functional consequences of these molecular and cellular abnormalities are examined with electrophysiological dissection of a somatosensory circuit and accompanied by deficits on a specific somatosensory discrimination task. This manuscript takes a significant step forward in understanding not only the roles of OPC synapses in postnatal development, but also in inhibitory neuronal maturation and function. For these reasons, the paper will be of significant interest to the broad fields of activity-dependent myelination and cortical inhibitory circuits.

1. Reduced firing frequency, synaptic connectivity with SSCs, and lower density of PV+ interneurons suggest that FSI maturation is delayed in these mutants. Are these effects a transient developmental delay in FSI maturation and myelination that is recovered during adulthood?

a-Are FSI density changes due to loss of PV expression, interneuron migration, or survival?

b-Previously the authors published that gamma2-GABAAR deletion in OPCs resulted in decreased

density of OPCs at P30 (Balía et al., 2017). How do these results reconcile with the current manuscripts findings of OPC density in Supp. Fig. 1. Along these lines, the authors should discuss recent findings of the relationship of interneuron and oligodendrocyte development (Voronova et al., 2017).

c-Changes in FSI maturation make ruling out changes in synaptic transmission difficult when modeling conduction velocity.

2. The authors suggest that “both axon morphology and myelin distribution were abnormal in the mutant.” (page 6, 2nd paragraph).

a-What are the morphology changes in the axon? Supp. Fig. 2 suggests interbranch point length and distance are unchanged. Other measurements of PV axonal arbor would be helpful in ascertaining the validity of this statement.

b-The author’s demonstrate that the distance to first MBP segment is increased in mutant, does this imply changes to the axon initial segment (AIS)? Since it is well known that changes to the AIS modulate excitability additional data (e.g. immunostaining) would strengthen the authors’ conclusions axon morphology is disrupted in the mutants.

3. In Fig. 3 the authors used Sholl analysis in biocytin-filled oligodendrocyte lineage cells (identified via inducible expression of GCaMP3 driven by NG2-CreER) to show that oligodendrocyte ramification is decreased in the gamma2 mutants. While the linear I-V curve is a parameter of mature oligodendrocytes (DeBiase et al., 2010; Kukley et al., 2010), immunostaining of mature oligodendrocyte marker expression would confirm that oligodendrocyte morphology is indeed aberrant in the mutants, and the results are not due to an imbalance of oligodendrocyte lineage progression. For example, if OPC differentiation was inhibited in the mutants, the probability of patching a premyelinating oligodendrocyte would be greater in the control animals, and thus would influence the mean # of branches.

4. Many mice were excluded from the behavioral analyses in Fig. 7 and in particular mutant mice (11/19 mutant mice). In the cited paper describing the behavioral test, the authors excluded only 2/18 mice for the same reasons. Can the authors comment on why the gamma2-f/f mice have such reduced exploratory behavior during the learning/testing phases? This seems contradictory to the open field behavior in Supp. Fig. 6. Additional clarity in the discussion of differences in exploration time would be helpful:

Results, page 14 “We observed that control mice were indeed able to discriminate between the two textures, as they spent significantly more time exploring the novel textured object compared to the old textured object they had already encountered (Fig. 7c).”

Results, page 15 “It is noteworthy to mention that both groups spent a similar amount of time exploring the textures, confirming that differences in novel texture exploration were due to the ability to discriminate and not a by-product of the amount of time spent exploring (exploration time: 9.74 ± 1.66 s for control mice, 9.06 ± 2.33 s for $\gamma 2f/f$ mice; $p=0.809$, two-sample Student’s t-test).”

Minor points

1-The discussion of Fig. 5g-k comes after Fig. 6 in results section, which is a bit confusing

2-There is a mismatch in the labeling and the legend of Supp. Fig. 1e.

3-It would be helpful to include the number of mice, in addition to number of cells for each experiment.

Reviewer #3 (Remarks to the Author):

Benamer et al investigate the consequences of disrupted GABAergic signaling onto oligodendrocyte progenitors (OPCs) in terms of fast-spiking interneuron (FSI) myelin and internode patterning, FSI electrophysiological properties/connectivity and whisker texture discrimination. Overall the authors find altered myelin patterning including longer internodes, longer nodes of Ranvier, and the presence of aberrantly myelinated axonal branches. Moreover, these neurons exhibit decreased firing frequency and disrupted conduction and/or disconnection with at least one population of cortical neurons. Finally, the authors use a whisker-based discrimination task to show that disrupting GABAergic receptor activation on OPCs results in a subtle defect in exploration behavior.

Overall the study is important for advancing our understanding of myelin structure and function in the cortex. However, it is somewhat incomplete in its current state. There are several aspects that should be expanded on in order to clarify the effect of this manipulation as outlined below.

1. The authors show that there is a difference in the distance to the first MBP signal from the cell soma of the FSI (Figure 1e). Does this correspond to a difference in length of the axon initial segment or is there an unmyelinated gap between the border of the AIS and the start of the MBP staining in the f/f mice? Does this at all relate or correlate to the increased node length in Figure 2?
2. Are there node of Ranvier proteins/channels present at the axon branch points that are covered by MBP in the f/f mouse? How does this compare to the distribution of these molecules at branch points in control mice?
3. Are there differences in axon diameter, length, or arborization for the FSI in the f/f mice?
4. The authors show that the nodes are longer in the f/f mice using Caspr staining only (Figure 3). Are there also changes in the distribution of voltage-gated channels at the nodes or are there regions of bare axon with no channels between the Caspr and Node channels?
5. The authors show that there are longer internodes and nodes but no change in the number of paranodes per micron (Figure 2d). This seems counterintuitive and requires an explanation.
6. Figure 3 suggests that mature(?) oligodendrocytes have a different morphology in the f/f mice. It seems that the authors used solely electrophysiology to determine the state of differentiation of these cells. Is this true? Based on the images provided these cells look to be in different stages of differentiation. Stage-specific markers should be used in addition to electrophysiology to be able to say that there is indeed a difference in cell morphology that is not due solely to these cells being OPCs vs premyelinating vs myelinating oligodendrocytes.
7. The authors describe the difference in PV + cells as an arrest of PV expression. Is this a change in the cell numbers or a change in the amount of PV expressed by individual cells? Is there more cell death of PV+ cells causing the difference in PV+ cell numbers or a decreased number of cells differentiating?
8. Figure 6 suggests an almost complete disconnection between the FSIs and the SSCs in the f/f mice

(Figure 6). This suggests not just a defect in conduction and strength of the FSI input to SSCs as shown in Figure 5 but instead no connection. Where are the IPSCs coming from in Figure 5 if there is a complete disconnection between FSIs and SSCs? Also, similar to question 3 above, what does the axonal arbor look like for the FSI at P30? If the authors label PV+ axons is there a complete lack of any PV+ axons in Layer IV?

9. Do any other neurons in the cortex display altered myelin profiles or is this defect specific to GABAergic FSIs?

Reviewer #1 (Remarks to the Author):

In the manuscript “Myelination of parvalbumin interneurons shapes the function of cortical sensory inhibitory circuits”, Benamer et al describe exciting new findings regarding the role of GABAergic neuron-glia signaling for myelination of axons of cortical interneurons. The authors report that reduction of GABAergic signaling between interneurons and oligodendrocyte progenitor cells, OPCs, (via deletion of gamma-2 subunit of GABAARs in OPCs) results in altered length of myelinated internodes and nodes of Ranvier in interneurons, changed firing pattern of interneurons with affected axonal myelination, and deficits in whisker-based texture discrimination. At present, mechanisms and functional significance of myelination of cortical interneurons during health and diseases remain poorly investigated. The study of Benamer et al sheds new light on the topic and inspires further research. I think that the study is well designed and carefully performed. It employs a combination of various approaches, from single-cell recordings to testing behavior of animals, and this design makes the conclusions solid. Yet I have few questions:

(1) The authors report that axons of interneurons in gamma-2fl/fl mice have longer nodes and internodes. Do the authors know whether the length of the axon of interneurons is altered in these mice? If the axonal length remains un-changed, longer nodes/ internodes probably indicate that fewer OLs are involved in myelination of each axon. And this in turn may mean that less OLs are generated (and are available for myelination) in gamma-2fl/fl mice, and/or that OLs in these mice are less eager to myelinate (because of e.g. energetic failure or other defects). Is it possible to reliably measure/estimate the length of the axons with altered myelination in gamma-2fl/fl mice, and also to count/estimate the number of OLs available to myelinate these axons? Estimating the length of the axon would also be interesting because the authors propose that FSI in gamma-2fl/fl mice may be less mature (page 9). May be maturation and axonal length are inter-related somehow?

We thank the reviewer for his/her very positive comments on our study. To answer the different questions asked by the three reviewers concerning axon morphology and length, we undertook a more in-depth analysis of biocytin-loaded layer IV FSI. We chose to recover all interconnected axon segments well beyond the myelinated region, while excluding disconnected segments that were the most distal and thinner processes. This procedure ensures that small dendritic branches are not included by mistake. It also reduces a possible variability due to the slicing and cell loading of thin processes. These new reconstructions allowed us to confirm that myelination of FSI axons is limited to the proximal axon region (Stedehouder et al., 2017, Cereb Cortex). We also corroborated that the lengths of the total reconstructed axon, the myelinated axon region and the myelinated and unmyelinated axon segments are equivalent between control and $\gamma 2^{fl/fl}$ mice (Supplementary Fig. 2c-f). Indeed, myelin of FSI is distributed in ~10% of our reconstructed axon segments in both groups and in ~2.5-4% when we consider the total axon length estimated for layer IV interneurons in the rat somatosensory cortex (Koelbl et al., 2015, Cereb Cortex). Furthermore, Sholl analysis revealed a very dense and ramified arborization that was similar between the two groups (Supplementary Fig. 2c). We concluded that the general complexity of the FSI axon is equivalent in control and $\gamma 2^{fl/fl}$ mice and that, as for other parameters such as the input resistance (which usually reflects morphology), the postnatal maturation of FSI axons is not interrupted in the mutant during postnatal development.

Unfortunately, technical limitations prevent the reliable estimation of the number of OLs that myelinate FSI axons. Although recombinant OL lineage cells express a fluorescent marker in control and $\gamma 2^{fl/fl}$ mice, the marker expression is often accumulated in the cytoplasm of the soma and more proximal branches, as often occurs in fluorescent transgenic lines. It is thus not possible to follow all myelinating processes of a single OL in these mice. It must also be considered that an OL can produce roughly 20 myelinating segments that would be intermingled with highly ramified dendrites and axons of FSI. This makes it difficult to simultaneously and unambiguously trace axonal segments myelinated by processes from identified OLs. Nevertheless, we know from our data that the length of the myelinated axonal region and the interbranch distances of myelinated segments of FSI axons are unchanged in the mutant (Supplementary Fig. 2d,f). Therefore, similar spatial rules govern segmental FSI myelination by OLs in control and $\gamma 2^{fl/fl}$ mice.

According to the aforementioned points, we modified Supplementary Figure 2 by adding the new data, and changed the text in Results, Discussion and Methods sections:

(P.6) « *The general morphology of FSI axons was assessed by 3D reconstructions where only clearly interconnected branches were taken into account. We found that the reconstructed axonal length as well as the complex axonal arborization, as showed by Sholl analysis, was unchanged between control and $\gamma 2^{fl/fl}$ mice (Supplementary Fig. 2c; reconstructed axonal length: $3565 \pm 437.7 \mu\text{m}$ for $n=8$ FSI in control mice and $3321 \pm 512.9 \mu\text{m}$ for $n=7$ FSI in $\gamma 2^{fl/fl}$ mice; $p=0.7789$, two-tailed Mann-Whitney U Test). As recently reported^{6,7,9}, all 3D reconstructions of recorded FSI were myelinated in both groups as revealed by the presence of internodes expressing the myelin basic protein (MBP) which were accommodated on only ~10% of the initial region of the reconstructed axon, and thus exhibited a biased distribution towards proximal axonal segments (Fig. 1a, b, Supplementary Fig. 2d). In addition, the mean interbranch axon length of the myelinated region was also unaffected in the mutant (Supplementary Fig. 2e, f). As previously reported⁹, short interbranch point segments were unmyelinated in the two groups (Supplementary Fig. 2f), suggesting that the same spatial rules govern segmental FSI myelination by OLs in control and $\gamma 2^{fl/fl}$ mice. »*

(P.10) « *While the general maturation in a complex and ramified axon did not appear to be affected in FSI of mutants, myelination defects in most if not all FSI caused an abnormal proximal axon morphology »*

(P.23) « *Considering that the total axon length of layer IV FSI in the rodent somatosensory cortex is comprised from 10 mm to 22 mm³³ and that FSI myelination is distributed within ~0.4 mm of the proximal axon (Supplementary Fig. 2d and Ref.⁹), only 2.5-4% of the total axon length of these interneurons is myelinated. Nevertheless, myelin anomalies of these interneurons cause dramatic changes in firing frequency, inhibitory circuit function and behavior. »*

(P.28) « *FSI axon morphology was determined from 3D reconstructions where only clearly interconnected branches were considered. Although this choice left aside the most distal and thinnest processes, it ensured that no small dendritic branch was included by error in the two groups. This procedure also reduces possible variations due to slicing, cell loading, and depth of recorded cells in slices, ensuring a more accurate comparison between groups. »*

(2) When the authors report the data regarding myelination defects in gamma-2fl/fl mice, they show that n = 8 cells (Figure 1) which may appear as a bit too low number. Do those

cells represent typical examples of how FSI look like in these mice or do they rather represent a sub-population? How easy/difficult is it to find the interneurons with altered myelination in gamma-2fl/fl mice? Are there also FSI with normal myelination in these animals? How easy is it to find those? How the number of FSI with altered myelination compares to the number of FSI with unchanged myelination? In addition, it is also not clear from how many animals the eight cells have been acquired? It is important to provide all these pieces of information in order to clarify for the reader whether the authors describe typical changes in the whole population of FSI, or whether they rather show the sub-population of cells in which myelination defects have occurred.

Biocytin-loaded FSI used for 3D reconstructions of the axon were randomly recorded in slices and a successful axon labeling was the only criterion to keep the cell for morphological analysis (one cell per slice ; n=8 FSI in N=5 control mice and n=9 FSI in N=3 $\gamma 2^{fl/fl}$ mice). As previously reported (Stedehouder et al., 2017, Cereb Cortex; Balia et al., 2017, Glia), we observed that the 17 recovered cells were myelinated, regardless of the mouse genotype. Nevertheless, contrary to what is suggested by the reviewer, there is no evidence for the existence of two FSI sub-populations in the mutant. In fact, the data in dot plots of Figure 1 and 2 do not appear to be distributed in separated groups (see also distributions in Fig. 1i and 2f) and, for most measured parameters, there is almost no overlap with the data in controls (see Fig. 1g, 1h, 2e and Supplementary Fig. 3b). Moreover, since full reconstructions require high resolution confocal images and several hours of careful inspection and analysis, we corroborated these data by looking at the initial part of FSI axons in PV cells randomly imaged in sections immunostained against PV and MBP (Supplementary Fig. 3a and 3b). We observed a similar defect in the proximal part of layer V FSI axons in the mutant (10 FSI in N=5 control mice and 10 FSI in N=6 $\gamma 2^{fl/fl}$ mice). Altogether, these findings allow us to conclude that reconstructed axons are representative of the FSI axon population rather than a sub-population, and that probably most if not all FSI had myelination defects in $\gamma 2^{fl/fl}$ mice.

To clarify all these points, we added:

(P.6) « *This was tested by comparing the axon morphology and myelin distribution of single biocytin-loaded FSI, randomly recorded, in control and $\gamma 2^{fl/fl}$ mice... »*

(P.7) « *...immunostained PV⁺ interneurons of layer V where both PV⁺ FSI and myelination are abundant¹⁹ (Supplementary Fig. 3a, b), corroborating that most FSI axons displayed this defect in the mutant. »*

(P.10) « *While the general maturation in a complex and ramified axons did not appear to be affected in FSI of mutants, myelination defects in most if not all FSI caused an abnormal proximal axon morphology »*

(P.28) « *Biocytin-loaded FSI were randomly selected during recordings. To avoid any biased during the analysis, a successful axon labeling was the only criterion to keep the cell for further analysis. »*

(3) Please, also indicate the number of animals AND the number of cells in all Figures (or Figure Legends).

We now included the number of animals used in each experiment in figure legends. We referred to the number of cells as n and to the number of animals as N. Note that the minimum number of animals per genotype and experiment was in most cases N=3 or more.

(4) The authors report that differentiated OLs in gamma-2fl/fl mice have altered morphology which they present in Figure 3. But are those cells really OLs? Why don't they have myelin sheathes?

As the question of the OL stage of morphologically altered cells was raised by the 3 reviewers, we undertook new experiments and analyses. We performed patch-clamp recordings of fluorescent layer IV OL lineage cells that were loaded with biocytin in order to correlate the profile of their I-V curves, the expression of OL lineage cell markers and their morphology. OPCs were recognized by a rectifying I-V curve, the presence of inward Na⁺ currents, an immunoreactivity for NG2 and a lack of expression of CC1. Morphological analyses revealed that OPCs did not display morphological differences between control and $\gamma 2^{fl/fl}$ mice (see new Supplementary Fig. 5). Conversely, OLs undergoing differentiation were recognized by a linear I-V curve, an immunoreactivity for CC1 and a lack of expression of NG2 (see new Fig. 3a,b). Although these cells were already engaged in a differentiation process, they never expressed MBP which allowed us to conclude that they consisted in premyelinating OLs. Unfortunately, we never recorded fully mature OLs in acute slices in either of the two mouse lines. In fact, premyelinating OLs already displayed a low GCamp3 fluorescence compared to OPCs under the microscope and were in general more difficult to find during recordings. It is thus possible that myelinating OLs were undetectable as the intrinsic fluorescence decreases along the maturation process. This difficulty was not encountered in immunostainings since the detection of the fluorescence is highly enhanced by GFP immunostainings as revealed by the presence of many GFP⁺/MBP⁺ processes in our samples. Despite the technical limitation during recordings, we were able to unambiguously determine that premyelinating OLs exhibit a simpler morphology in $\gamma 2^{fl/fl}$ mice than in controls (Fig. 3c-e). Interestingly, our new results suggest that the inactivation of OPC GABAergic synapses mainly affects the transition from a progenitor state to an OL state. To clarify this point, we were more precise on the designation of the recorded OL lineage cells throughout the text and included a new Supplementary Fig. 5, a new Figure 3a,b and added in the results and discussion section:

(P9) « Severe myelination defects and axon malformation of FSI most probably resulted from altered OL lineage cells in layer IV. To test this possibility, we analyzed the morphology of biocytin-loaded recombinant cells recognized by their fluorescence during patch-clamp recordings at P26-P35. Among the recorded cells, we identified OPCs by a rectifying I-V curve, the presence of inward Na⁺ currents and an immunoreactivity for NG2²². (Supplementary Fig. 5a, b). We also found that all recorded OPCs lacked the OL marker CC1 (not shown). Morphometric measurements using Sholl analysis did not reveal any difference in the morphological complexity of OPC processes between control and $\gamma 2^{fl/fl}$ mice (Supplementary Fig. 5c-f). In comparison, OLs undergoing differentiation were identified by a linear I-V curve and an immunoreactivity for CC1 (Fig. 3a,b). Although we tested in four occasions that these cells lacked the OPC marker NG2, none of these cells expressed MBP, even at P84-P93, indicating that they were in a premyelinating OL state in both groups (not shown; n=12 and n=11 for N=6 control and N=6 $\gamma 2^{fl/fl}$ mice, respectively). It is noteworthy that Gcamp3 fluorescence decreases with the maturational state of the cells in acute slices of these mice, which prevented us from detecting myelinating OLs during recordings. Nevertheless, morphometric measurements of biocytin-loaded premyelinating OLs revealed that the number

of branch intersections from the soma was reduced in $\gamma 2^{ff}$ mice compared to controls (Fig. 3c-e-). »

(P19) « Interestingly, the reduced morphological complexity of premyelinating OLs in the mutant suggests that the inactivation of OPC GABAergic synapses mainly affects the transition from a progenitor to an OL state. »

(5) The authors present the E/I ratio and some alterations in it. These experiments are interesting, but they are performed in whole-cell mode with « artificial » chloride concentration. Hence, it is not clear how these findings reflect the real situation regarding the balance between excitation and inhibition in vivo, where chloride concentration in interneurons may be different from the one in the recording pipette, and it may also depend on the age, and it may also be altered in gamma-2fl/fl mice (it is clear that only GABARs in OPCs are expected to be affected, but a feed-back mechanism from myelinating cells to neurons may exist). Please, explain/comment on this issue.

In the cerebral cortex, GABA switches from a depolarizing to a hyperpolarizing neurotransmitter around P8 (Allène et al., 2008, J Neurosci), when there are no obvious defects in $\gamma 2^{ff}$ mice. Indeed, at P10, PV⁺ cell density as well as proliferation, differentiation and cell density of OPCs are similar between controls and mutants (Fig. 4e, 4f, Supplementary Fig. 1f-i and Balia et al., 2017, Glia). Furthermore, the switch from depolarizing to hyperpolarizing in GABA in neurons results from a reduction in the intracellular chloride concentration which is accompanied by an increased expression of the potassium/chloride co-transporter KCC2. The expression of this co-transporter, which is sufficient to end the depolarizing period of immature cortical neurons (Lee et al., 2005, EJM), occurs in the somato-dendritic compartment rather than in the axon (Côme et al., 2019, Front Cell Neurosci). Finally, we reported major defects of IPSCs and E/I ratio in excitatory spiny stellate cells (SSCs; Fig. 5), but not in FSI (Supplementary Fig. 8). For all these reasons, it is unlikely that impaired PV interneuron-oligodendroglia interactions specifically revert the inhibitory effect of GABA in SSCs at P30. To take in consideration this reviewer comment, we added in the discussion:

(P.22) « Since GABA switches from a depolarizing to a hyperpolarizing neurotransmitter around P8 in the cerebral cortex⁴⁸ and as reduced IPSCs occurred in SSCs, but not FSI, at a later developmental stage in $\gamma 2^{ff}$ mice, it is very likely that a decreased FSI-mediated inhibition exists in SSCs in vivo in the mutant. »

(6) When the authors test for relationship between altered firing of FSI and myelination impairments, they analyze electrophysiological properties of spiny stellate cells for comparison. But those cells are excitatory neurons (to my knowledge). What is the rationale for using SSCs for comparison in the described experiments? Would not it be more logical to compare the firing pattern of FSIs with affected myelination to the firing pattern of other cortical interneurons whose myelination is (presumably) not affected in gamma-2fl/fl mice?

We compared the electrophysiological properties of FSI with those of SSCs because these two cell types are the main neurons involved in layer IV thalamocortical circuits. Our results showed that the intrinsic electrophysiological properties selectively changed in FSI in these circuits. We did not compare these properties with those of other interneurons because it is known that non-fast-spiking interneurons are not or barely myelinated (Micheva et al., 2016, eLife; Stedehouder et al., 2017, Cereb Cortex). However, we agree that this point needed further clarification. We thus decided to analyze the electrophysiological properties of layer V pyramidal neurons that

are known to be highly myelinated in the cerebral cortex. We did not find differences in their firing properties in the mutant, strengthening our results showing that these defects are FSI-specific (See Supplementary Table 1). To clarify this point, we added:

(P.11) « *Finally, we also analyzed the electrophysiological properties of layer V pyramidal neurons, a cell type known to be highly myelinated in the cerebral cortex, and did not find any difference either (Supplementary Table 1). These data indicate that the abnormal axon morphology and myelination of FSI changed the intrinsic excitability of these interneurons but not that of SSCs or pyramidal cells.* »

(7) The Discussion part is presented largely as expanded repetition of the Results section. I think it could be shortened. And it would be interesting to include some thoughts or inspiring ideas for future research. For instance: What could be the mechanisms through which axon-OPC signaling regulates/modulates firing pattern of interneurons? Is it a kind of feed-back mechanism? Signaling from axons to the soma? Can it be that longer nodes of Ranvier attract more Na-channels to the axon, and this in turn results in diminished number of Na-channels in the neuronal soma and in the axon-hillock, and hence in altered firing? What is already known regarding alterations of myelination of interneurons during diseases? How those alterations compare to the present findings of the authors?

We thank the reviewer for allowing us to improve the discussion that was indeed repetitive. We now removed several sentences and discussed new points that allowed us to consider the questions mentioned above and some other comments raised by reviewer 2 and 3 (see Discussion section). Since the decrease in the high firing frequency of FSI in $\gamma 2^{fl/fl}$ mice was not accompanied by modifications of other important electrophysiological properties such as input resistance, resting potential, AHP, action potential amplitudes and durations (Supplementary Table 1), our results favored the idea that the reduction in the firing frequency is mainly caused by a reduced conduction rather than by changes in channel expression or potential feedback mechanisms between axons and OPCs that should occur at early developmental stages (see P. 19 and 20). Regarding alterations of myelination of interneurons during diseases, we added at the end of the discussion:

(P. 23) « *Interestingly, FSI constitute a recurrent locus of dysfunctions in neurodevelopmental diseases such as schizophrenia. Notably, the synchronization of neuronal ensembles in the gamma range frequency, which largely depends on FSI activity⁵⁰, is commonly altered in this disease. Moreover, a recent study performed in a rat model of schizophrenia showed a hypomyelination of cortical PV⁺ interneurons⁵¹. Considering our demonstration of the important role of FSI myelination in regulating the function of FSI, it is thus possible that FSI myelination defects alter local cortical circuit oscillations in vivo which, in turn, contribute to cognitive deficits observed in this disease^{5,15}.* »

Reviewer #2 (Remarks to the Author):

Benamer et al. present an intriguing study that provides novel insights into the roles of GABAergic OPC synapses and myelination in cortical circuit development. They combine electrophysiological, post-hoc immunohistochemical, and behavioral methods to establish distinct morphological and behavioral phenotypes associated with oligodendrocyte lineage deletion of the gamma-2 subunit of GABAARs. Furthermore, this manuscript aims to establish a bidirectional signaling mechanism, through which fast-spiking interneurons (FSI) -> OPC synaptic input regulates not only the morphology of oligodendrocytes and myelination of FSI, but also the axonal morphology and number of PV-positive interneurons in the cortex. The functional consequences of these molecular and cellular abnormalities are examined with electrophysiological dissection of a somatosensory circuit and accompanied by deficits on a specific somatosensory discrimination task. This manuscript takes a significant step forward in understanding not only the roles of OPC synapses in postnatal development, but also in inhibitory neuronal maturation and function. For these reasons, the paper will be of significant interest to the broad fields of activity-dependent myelination and cortical inhibitory circuits.

1. Reduced firing frequency, synaptic connectivity with SSCs, and lower density of PV+ interneurons suggest that FSI maturation is delayed in these mutants. Are these effects a transient developmental delay in FSI maturation and myelination that is recovered during adulthood?

a-Are FSI density changes due to loss of PV expression, interneuron migration, or survival?

To answer this question, we performed a new set of immunostainings against PV and the general neuronal marker NeuN in control and $\gamma 2^{ff}$ mice at P120 to determine if the decrease in PV⁺ cells in the mutant was transient during postnatal development or persisted during adulthood. We found that neither the density of PV⁺ cells nor PV-cell/NeuN-cell ratio was changed in the mutant at P120 (new Fig. 4f and Supplementary Fig. 6). These results suggest a delayed expression of the PV protein in $\gamma 2^{ff}$ mice, during the period of active myelination in the cortex, rather than a compromised survival of PV⁺ neurons. To clarify this point, we added in the results section:

(P.11) « *To assess potential FSI developmental impairments in $\gamma 2^{ff}$ mice, we analyzed the density of PV cells at P10, P24, P30 and P120 (Fig. 4e,f). Although we observed the expected increase in PV cells from P10 to P30 in control mice, their density remained stable during the first postnatal month in the mutant, after which it only increased at P120 (Fig. 4e,f). Neither the density of PV⁺ neurons nor the ratio PV⁺ cells/NeuN⁺ cells changed during adulthood, suggesting a delayed expression of the PV protein rather than a loss of PV cells in $\gamma 2^{ff}$ mice (Fig. 4f and Supplementary Fig. 6). In addition, while no changes were detected at P10 and P120 between control and $\gamma 2^{ff}$ mice, significant differences were observed at P24 and P30, indicating that the early disruption of FSI-OPC interactions resulted in a PV expression deficiency during the developmental myelination process (Fig. 4f).* »

b- Previously the authors published that gamma2-GABAAR deletion in OPCs resulted in decreased density of OPCs at P30 (Balía et al., 2017). How do these results reconcile with the current manuscripts findings of OPC density in Supp. Fig. 1. Along these lines, the authors should discuss recent findings of the relationship of interneuron and oligodendrocyte development (Voronova et al., 2017).

We thank the reviewer for allowing us to clarify this point that was not properly explained in the manuscript. In Balía et al. (2017), we showed that GABAergic synapses of OPCs does not impact the proliferation and differentiation of these progenitors during postnatal development. We found that OPC and OL densities were similar between controls and $\gamma 2^{ff}$ mice at different developmental stages, except at P30 where a small but significant OPC density decrease was observed. In line with Balía et al. (2017), we reported here a robust decrease of GABAergic synaptic activity at P10 in $\gamma 2^{ff}$ mice was not accompanied by changes in layer IV OPC density (Fig. 3a-e in Balía et al. vs. our present Supplementary Fig. 1). This developmental stage is the most relevant to validate the model since it corresponds to the peak of OPC synaptic connectivity (Orduz et al., 2015, eLife). We thus did not reproduce all time points reported in Balía et al. (2017) to avoid being redundant. Overall, our present findings together with those described in Balía et al. (2017) point to a major role of FSI-OPC synapses in guiding the correct FSI myelination rather than in regulating OPC development. To clarify all these points and discuss Voronova et al., (2017), we added in the discussion:

(P.19) « *This interdependency probably already exists early on during postnatal development as FSIs synaptically contact OPCs during the first two postnatal weeks^{8,14}. Although $\gamma 2$ -mediated GABAergic synapses of OPCs do not impact the proliferation and differentiation of these progenitors at P10 (Supplementary Fig. 1 and Ref.⁶), the early inactivation of the $\gamma 2$ subunit of GABA_A receptors in these progenitors induces long-term adverse effects on oligodendroglia by reducing OPC density⁶ and causing an aberrant morphology of premyelinating OLs in the mature cortex (Fig. 3). Interestingly, the reduced morphological complexity of premyelinating OLs in the mutant suggests that the inactivation of OPC GABAergic synapses mainly affects the transition from a progenitor to an OL state. In turn, we observed an aberrant axon morphology and myelination of FSI. The idea that interneurons and oligodendroglia are reciprocal partners during development is reinforced by recent data showing that 1) both cell types are born from progenitors expressing similar transcription factors and lying in the same germinal regions^{8,38,39}; 2) lineage-related interneurons and OPCs are initially over-produced and then significantly demised at early postnatal stages^{8,38,40}; 3) surviving lineage-related interneurons and oligodendroglia form anatomical and functional clusters at postnatal stages⁸ and 4) migrating interneurons secrete the cytokine fractaline which promotes oligodendrogenesis via the fractaline receptor CX3CR1 expressed in OPCs⁴¹ »*

c- Changes in FSI maturation make ruling out changes in synaptic transmission difficult when modeling conduction velocity.

We agree that we cannot totally rule out that synaptic changes contribute to the delay of IPSC responses of SSCs in $\gamma 2^{ff}$ mice. However, it is known that synaptic latencies dependent on release mechanisms vary in an amplitude-dependent manner (Boudkkazi et al., 2007, Neuron) which is not the case in our study (see P. 14). Our experimental data thus suggest that the latency delay is probably mainly caused by an alteration of the conduction velocity rather than a deficit in neurotransmission. Moreover, computational simulations aim to predict the speed at which action potentials are propagated in an axon with specific myelin properties, a

characteristic that is independent of synaptic inputs (Arancibia-Carcamo et al., 2017, eLife). However, we understand that we need to be more careful and precise about our conclusions and we thus added:

(P. 14) « *Although we cannot totally rule out that neurotransmission deficits of presynaptic FSI contributed to the latency delay of IPSC responses in the mutant, the experimental latency retardation -that is independent of IPSC amplitudes- and the slow conduction velocity obtained by computational simulations support the fact that the abnormal longer nodes and internodes of FSI affects their ability to rapidly conduct action potentials.* »

2. The authors suggest that both axon morphology and myelin distribution were abnormal in the mutant (page 6, 2nd paragraph).

a-What are the morphology changes in the axon? Supp. Fig. 2 suggests interbranch point length and distance are unchanged. Other measurements of PV axonal arbor would be helpful in ascertaining the validity of this statement.

To answer this question, we further reconstructed and analyzed biocytin-loaded FSI axons (please refer to question 1, reviewer 1, for details). We found no changes in the dense axonal arborization or myelinated axon length of FSI between control and $\gamma 2^{f/f}$ mice. Major defects remained restricted to the proximal region of the axon.

b-The authors demonstrate that the distance to first MBP segment is increased in mutant, does this imply changes to the axon initial segment (AIS)? Since it is well known that changes to the AIS modulate excitability additional data (e.g. immunostaining) would strengthen the authors' conclusions axon morphology is disrupted in the mutants.

To answer this question, we performed immunostainings against PV and Ankyrin G, a recognized marker of the axon initial segment (AIS). We found that the length and location of the AIS in PV⁺ axons remained unchanged between control and $\gamma 2^{f/f}$ mice (see new Supplementary Fig. 4). This observation does not invalidate our previous findings since we also observed that the axonal region from the soma to the first branching point was longer and accommodated more internodes in the mutant (Fig. 1f and 1g). Although defects appeared in the proximal region, they extended beyond the beginning of the axon. To clarify this point, we added a new Supplementary Fig. 4 and a text in the results section:

(P.7) « *However, the elongation of the initial part of the axon in $\gamma 2^{f/f}$ mice was not accompanied by an increased length or a displacement of PV⁺/Ankyrin G⁺ axon initial segment (AIS) (Supplementary Fig. 4a,b).* »

3. In Fig. 3 the authors used Sholl analysis in biocytin-filled oligodendrocyte lineage cells (identified via inducible expression of GCaMP3 driven by NG2-CreER) to show that oligodendrocyte ramification is decreased in the gamma2 mutants. While the linear I-V curve is a parameter of mature oligodendrocytes (DeBiase et al., 2010; Kukley et al., 2010), immunostaining of mature oligodendrocyte marker expression would confirm that oligodendrocyte morphology is indeed aberrant in the mutants, and the results are not due to an imbalance of oligodendrocyte lineage progression. For example, if OPC differentiation was inhibited in the mutants, the probability of patching a premyelinating oligodendrocyte would be greater in the control animals, and thus would influence the mean # of branches.

We thank all reviewers for this constructive comment that we fully answered above (please refer to question 4, reviewer 1, for details). We determined that the cells displaying a less complex morphology in $\gamma 2^{f/f}$ mice were premyelinating OLs and not OPCs (see new Supplementary Fig.5, new Figure 3a, 3b and P.9 and P.19).

4. Many mice were excluded from the behavioral analyses in Fig. 7 and in particular mutant mice (11/19 mutant mice). In the cited paper describing the behavioral test, the authors excluded only 2/18 mice for the same reasons. Can the authors comment on why the gamma2-f/f mice have such reduced exploratory behavior during the learning/testing phases? This seems contradictory to the open field behavior in Supp. Fig. 6. Additional clarity in the discussion of differences in exploration time would be helpful:

Results, page 14 1C; We observed that control mice were indeed able to discriminate between the two textures, as they spent significantly more time exploring the novel textured object compared to the old textured object they had already encountered (Fig. 7c);

Results, page 15 1C; It is noteworthy to mention that both groups spent a similar amount of time exploring the textures, confirming that differences in novel texture exploration were due to the ability to discriminate and not a by-product of the amount of time spent exploring (exploration time: 9.74 ± 1.66 s for control mice, 9.06 ± 2.33 s for $\gamma 2^{f/f}$ mice; $p=0.809$, two-sample Student 19; s t-test).

We would like to thank the reviewer for highlighting this point which was unclear in the manuscript. Some mice were indeed excluded from the analysis based on a lack of active exploration of the textured objects (lesser than 2 seconds) during either the learning or testing phase, as described previously in the literature (Wu et al., 2013). This exclusion criterion reflects a lack of interest for the textured objects rather than a lack of overall exploratory behaviour in the arena, since similar levels of exploration were observed in the open field (Supplementary Fig. 9). We have thus modified the manuscript to make this point clearer in the text:

(P.16) « For analysis, we considered only mice which explored the two objects for more than 2 s during the learning and testing phases, as previously described³⁶. We found that 4 out of 14 control mice and 11 out of 19 $\gamma 2^{f/f}$ mice lacked sufficient exploration of the textured objects and were thus excluded. To test whether this insufficient object exploration was due to a lower overall exploratory behavior in the arena, we tested the exploratory behavior in a conventional open-field. We found no significant differences between control and $\gamma 2^{f/f}$ mice on the distance and time spent in the outer and inner zones, the total time of activity and inactivity and the time course of traveled distance (Supplementary Fig. 9). The lack of object exploration was therefore most likely due to a lack of interest for the objects rather than a deficient exploratory capacity. In addition, we did not observe significant differences in whisker-based texture exploration during the learning phase between control and $\gamma 2^{f/f}$ mice, which ensures comparable levels of exploratory behavior among mice included in the analysis (Fig. 7b). We then checked (...) It is worthy to mention that mice from both groups considered for analysis spent similar amounts of time exploring the textures during the testing phase, confirming that differences in novel texture exploration were due to the ability to discriminate and not a by-product of the amount of time spent exploring the objects (exploration time: 9.74 ± 1.66 s for control mice, 9.06 ± 2.33 s for $\gamma 2^{f/f}$ mice; $p=0.809$, two-sample Student's t-test). »

Minor points

1-The discussion of Fig. 5g-k comes after Fig. 6 in results section, which is a bit confusing

To clarify this point, we moved the description of Fig. 6 below that of Fig. 5g-k (P.15).

2-There is a mismatch in the labeling and the legend of Supp. Fig. 1e.

We now corrected the mismatch in the Supp Fig. 1e. We also replaced the plots of the percentage of cells by the densities of cells in Supp. Fig. 1h, 1i to better fit with the description given in the first paragraph of results in the main manuscript (P. 6) and in the Supp. Figure legend.

3-It would be helpful to include the number of mice, in addition to number of cells for each experiment.

As mentioned for reviewer 1, we now included the number of mice (N) for each experiment in figure legends.

Reviewer #3 (Remarks to the Author):

Benamer et al investigate the consequences of disrupted GABAergic signaling onto oligodendrocyte progenitors (OPCs) in terms of fast-spiking interneuron (FSI) myelin and internode patterning, FSI electrophysiological properties/connectivity and whisker texture discrimination. Overall the authors find altered myelin patterning including longer internodes, longer nodes of Ranvier, and the presence of aberrantly myelinated axonal branches. Moreover, these neurons exhibit decreased firing frequency and disrupted conduction and/or disconnection with at least one population of cortical neurons. Finally, the authors use a whisker-based discrimination task to show that disrupting GABAergic receptor activation on OPCs results in a subtle defect in exploration behavior.

Overall the study is important for advancing our understanding of myelin structure and function in the cortex. However, it is somewhat incomplete in its current state. There are several aspects that should be expanded on in order to clarify the effect of this manipulation as outlined below.

1. The authors show that there is a difference in the distance to the first MBP signal from the cell soma of the FSI (Figure 1e). Does this correspond to a difference in length of the axon initial segment or is there an unmyelinated gap between the border of the AIS and the start of the MBP staining in the f/f mice? Does this at all relate or correlate to the increased node length in Figure 2?

We thank the reviewer for this comment that was also raised by reviewer 2 (please refer to question 2b of reviewer 2 for details). We found that the difference in the distance to the first MBP signal from the soma does not correspond to changes in the AIS since the length and location of PV⁺/Ankyrin G⁺ axon segments were equivalent between control and $\gamma 2^{f/f}$ mice and (see new Supplementary Fig. 4). Furthermore, we did not observe any correlation between the node length and the axonal length to the first MBP signal or to the first branch point in controls and $\gamma 2^{f/f}$ mice. To clarify this last point, we added in the results section:

(P. 8) « *It is noteworthy that we did not find any correlation between the mean node length and the axonal length to the first MBP signal or to the first branch point in controls and $\gamma 2^{f/f}$ mice, suggesting that the length of these different axonal subregions are most likely independently regulated (correlation coefficient of 0.250, $p=0.595$ in controls and of 0.336, $p=0.882$ in $\gamma 2^{f/f}$ mice for the correlation of node length with length to the first MBP signal; correlation coefficient of -0.498, $p=0.256$ in controls and of 0.005, $p=0.991$ in $\gamma 2^{f/f}$ mice for correlation of node length with length to the first branch point; Spearman correlation). »*

2. Are there node of Ranvier proteins/channels present at the axon branch points that are covered by MBP in the f/f mouse? How does this compare to the distribution of these molecules at branch points in control mice?

To answer this question, we examined whether MBP co-stained with Caspr at branch points of labeled biocytin-loaded FSI axons. In control mice, branch points of myelinated axons were

often delimited by a Caspr signal present at the extremity of MBP⁺ internodes (branch points remaining naked). In $\gamma 2^{f/f}$ mice, however, we observed a lack of Caspr expression at the same level in aberrantly myelinated branch points. Therefore, this first approach did not reveal any indication for the expression of proteins/channels of nodes of Ranvier at the level of branch points covered by myelin in the mutant. Unfortunately, the proteins/channels present at branch points has been in general less described than those at nodes of Ranvier. Furthermore, the protein/channel composition of nodes in FSI axons is also unknown (see also our answer to question 4). We believe that the study of the identity and distribution of proteins/channels in branch points and nodes of Ranvier of FSI axons constitutes a broad topic for future studies, outside the scope of the present work. However, to take into consideration the reviewer's comment, we mentioned these points in the results sections as follows:

(P.8) « *Moreover, the Caspr protein, detected in immunostainings and normally expressed at the extremity of MBP⁺ internodes facing branch points, was never expressed at this axonal site in abnormally myelinated ramifications of FSI in $\gamma 2^{f/f}$ mice (not shown), suggesting that branch points aberrantly covered by myelin may lack proteins present in nodes of Ranvier.* »

3. Are there differences in axon diameter, length, or arborization for the FSI in the f/f mice?

As mentioned for reviewer 1 and 2, we answered this question by further reconstructing biocytin-loaded FSI axons (please refer to point 1, reviewer 1, to get more details concerning axon morphology). We performed Sholl analysis and found no changes in the dense axonal arborization of FSI between control and $\gamma 2^{f/f}$ mice. Moreover, the reconstructed axon length and the myelinated axon length were equivalent in the two groups.

4. The authors show that the nodes are longer in the f/f mice using Caspr staining only (Figure 3). Are there also changes in the distribution of voltage-gated channels at the nodes or are there regions of bare axon with no channels between the Caspr and Node channels?

We agree with the reviewer that defining the distribution of voltage-gated channels at nodes of Ranvier of PV interneuron axons is a very interesting point. Indeed, myelination of this important neuronal subtype is a recently growing topic and many aspects remain to be discovered. However, for this same reason, available information on the molecular identity of the channels expressed at their nodes in normal conditions is limited. It must be considered that PV interneurons display a myelination pattern, axonal molecular properties and a node structural organization distinct from those of excitatory neurons (Micheva et al., 2016, eLife; Micheva et al., 2018, eNeuro). Thus, we would also expect major differences on channel expression and distribution at the level of nodes of Ranvier. We thus believe that a complete study on this question is necessary, independently of the present work. Nevertheless, we can partially answer the reviewer's comment with our electrophysiological data. While the firing frequency of PV⁺ FSI is significantly reduced in $\gamma 2^{f/f}$ mice (Fig. 4a, 4c and Supplementary Fig. 3c-e), all other properties such as the threshold, large amplitude and narrow duration of action potentials, large AHP and low input resistance remain unchanged (Supplementary Table 1). These results suggest that no major changes in the expression of channels occurred along PV interneuron of the mutant. We now discussed these points in the discussion section:

(P.20) « *In fact, myelination defects of FSI caused a significant reduction in their high firing frequency without modifying other properties such as input resistance, resting potential, AHP, action potential amplitudes and fast spike kinetics. This lack of changes in intrinsic electrophysiological properties makes major modifications in protein and channel expression at the AIS and nodes of Ranvier in the mutant rather unlikely. Future studies, however, will be needed to determine the molecular identity and distribution of different proteins and ion channels in myelinated FSI axons both in normal and pathological conditions. Nevertheless, although an increase in internode length tends to increase action potential conduction⁴⁵, aberrant longer internodes and nodes like those observed in $\gamma 2^{f/f}$ mice result in a decreased predicted conduction velocity (Fig. 5k). The slow conduction could thus be the main cause of the reduced firing frequency of FSI in the mutant.* »

5. The authors show that there are longer internodes and nodes but no change in the number of paranodes per micron (Figure 2d). This seems counterintuitive and requires an explanation.

We would like to thank the reviewer for raising this point that we overlooked. By verifying the raw data used to construct the bar plots of the previous manuscript, we realized that some values of paranodes/ μm in the summary table used to do the plots were incorrect. We now corrected this mistake and found that the number of paranodes was indeed significantly reduced in $\gamma 2^{f/f}$ mice ($p=0.021$, two-tailed Mann-Whitney U test). We apologize for this mistake that, fortunately, does not impact other results. We now rechecked all values and tables used in our study and did not find other discrepancies. We also provide a source table containing data used for all figures and tables and corrected Figure 2d and the text:

(P.8) « *Remarkably, we found an increased mean node length per cell accompanied by a decreased number of paranodes per μm in $\gamma 2^{f/f}$ mice (Fig. 2d,e).* »

6. Figure 3 suggests that mature(?) oligodendrocytes have a different morphology in the f/f mice. It seems that the authors used solely electrophysiology to determine the state of differentiation of these cells. Is this true? Based on the images provided these cells look to be in different stages of differentiation. Stage-specific markers should be used in addition to electrophysiology to be able to say that there is indeed a difference in cell morphology that is not due solely to these cells being OPCs vs premyelinating vs myelinating oligodendrocytes.

We thank all reviewers for this constructive comment that we fully answered above (please refer to question 4, reviewer 1, for details). We determined that the cells displaying a less complex morphology in $\gamma 2^{f/f}$ mice were premyelinating OLs and not OPCs (see new Supplementary Fig.5, new Figure 3a, 3b and P.9 and P.19).

7. The authors describe the difference in PV + cells as an arrest of PV expression. Is this a change in the cell numbers or a change in the amount of PV expressed by individual cells? Is there more cell death of PV+ cells causing the difference in PV+ cell numbers or a decreased number of cells differentiating?

As we answered to reviewer 2 (please refer to question 1a of this reviewer), our new results on PV and NeuN labelling at P120 suggest a delayed expression of the PV protein in $\gamma 2^{f/f}$ mice rather than neuronal death. We clarified this point in new Fig. 4f, Supplementary Fig. 6 and the result section (P.11).

8. Figure 6 suggests an almost complete disconnection between the FSIs and the SSCs in the f/f mice (Figure 6). This suggests not just a defect in conduction and strength of the FSI input to SSCs as shown in Figure 5 but instead no connection. Where are the IPSCs coming from in Figure 5 if there is a complete disconnection between FSIs and SSCs? Also, similar to question 3 above, what does the axonal arbor look like for the FSI at P30? If the authors label PV+ axons is there a complete lack of any PV+ axons in Layer IV?

We thank the reviewer to allow us to clarify this point that was not discussed in the previous manuscript. In our study, we performed paired recordings between two very close FSI and SSC to ensure that we stayed within a single barrel and in the area of the myelinated part of the FSI axon (<40 μm intersomatic distances). Although it is true that we did not find connected pairs in $\gamma 2^{\text{f/f}}$ mice, the FSI axon also projects towards adjacent barrels and layers to a lesser extent. The reduced current evoked by electrical thalamic stimulation in the mutant may thus result from distal FSI axon-to-SSC connections. In addition, it is possible that few somatostatin-expressing interneurons participate to feedforward inhibition at P30 (Tuncdemir et al., 2016, Neuron), contributing to the remained evoked current in the mutant. As mentioned before, the distribution or ramification of the FSI axon in $\gamma 2^{\text{f/f}}$ mice were unchanged and thus modifications of the axonal arbor cannot explain the observed lack of connectivity. To clarify this point, we added in the discussion:

(P. 21) « *While the feedforward inhibition of SSC was significantly reduced in the mutant upon electrical thalamic stimulation, our paired recordings reveal a lack of FSI-SSC connectivity within a single barrel and in the area of the myelinated part of the FSI axon (i.e. <40 μm intersomatic distances). Distal FSI (e.g. FSI in adjacent barrels or layers) as well as some somatostatin-expressing interneurons⁴⁸ may participate to the reduced IPSCs evoked by electrical thalamic stimulation at P30 in the mutant.* »

9. Do any other neurons in the cortex display altered myelin profiles or is this defect specific to GABAergic FSIs?

Considering that axons from many myelinated intracortical, intercortical and subcortical neurons are found in the cortex, it is very difficult to answer this question. However, although we cannot totally rule out that myelination defects occur in other neurons in $\gamma 2^{\text{f/f}}$ mice, we found that, unlike FSI, the electrophysiological properties of layer IV SSC as well as those of layer V pyramidal cells, a cortical neuron subtype known to be highly myelinated, remained unchanged (please see the new Supplementary Table 1 and our answer of point 6, reviewer 1). In addition, the amplitudes and latencies of EPSCs evoked in SSCs by direct thalamic inputs remained the same in the mutant while those of FSI-mediated IPSCs are reduced (Fig. 5). Thus, if myelination profiles are modified in other neurons in the mutant, they do not appear to significantly interfere with their physiological properties. To clarify this point, we added in the Discussion section:

(P.20) « *Finally, although we cannot totally rule out that myelination defects occur in other neurons in the cortex of $\gamma 2^{\text{f/f}}$ mice, we did not observe changes in the electrophysiological properties of SSCs and pyramidal neurons as well as in the amplitudes and latencies of EPSCs evoked in SSCs by direct thalamic inputs (while those of FSI-mediated IPSCs were reduced). Thus, if myelination profiles are modified in other neurons in the mutant, these changes do not appear to significantly interfere with their physiological properties.* »

Reviewer #1 (Remarks to the Author):

In the manuscript with the title “Myelination of parvalbumin interneurons shapes the function of cortical sensory inhibitory circuits”, the team of Maria-Cecilia Angulo demonstrated that genetic disruption of GABAergic synaptic signalling between fast-spiking interneurons (FSI) and oligodendrocyte precursor cells results in severe FSI myelination defects later in development. FSI with altered axonal myelination showed reduction in high-frequency firing and weaker connectivity with excitatory cortical neurons. Feedforward inhibition performed by FSI was also reduced. These alterations were accompanied by diminished whisker-dependent texture discrimination in the behavioural tests.

In the revised version of their manuscript, the authors addressed the comments of the reviewers by performing additional experiments, including new pieces of data, re-arranging the text of the manuscript, and improving the Discussion section. I think that the manuscript has become clearer and more complete now, and appears stronger than the original version.

The major strengths of this study are the novelty and the combination of multiple state-of-the-art approaches (including electrophysiology, imaging, modelling, behavioural paradigms, and transgenic animals) used to obtain the results. The findings of this study are clearly novel, and are expected to be of interest for neuronal and glial physiologists, as well as for clinical scientists. Neuronal physiologists rarely consider functional significance of myelination for proper function of neuronal circuits, and the present study strongly emphasizes this important point. Glial physiologists are still confused about possible functional role of axon-glia synapses, and the present study shows that GABAergic axon-glia signalling is involved in regulation of myelination of interneurons. Finally, in the recent years new evidence regarding the role of myelination in schizophrenia has been emerging, and the findings of the present study suggest that disturbance of GABAergic axon-glia signalling may be a key mechanism to consider when studying myelination defects in psychiatric (and perhaps other neurological) disorders. With this in mind, it is very likely that the study of Benamer and colleagues will be highly cited and will pave the way to new future discoveries in the field.

Reviewer #2 (Remarks to the Author):

The authors have thoroughly addressed my concerns with the addition of new data and discussion of their results. Their manuscript will push forward the field's understanding of neuron-OPC interactions and how they influence the functional maturation of fast spiking interneurons in cortical circuits.

Reviewer #3 (Remarks to the Author):

Overall the authors have addressed many of my previous points however one remaining concern is the analysis of the oligodendrocyte morphology in Figure 3. The authors use CC1 staining and electrophysiology to state that the effect is exclusive to premyelinating cells. As stated before the images of the biocytin filled cells look very reminiscent of different stages of differentiation. Since

CC1 is expressed early on during differentiation and continues even in mature cells I don't feel that the authors can conclusively state that premyelinating oligodendrocytes have an aberrant morphology. In my opinion one could just as easily conclude that the cells are in a different stage of differentiation so there might be a shift in the predominant population in the f/f mice. The authors used MBP to conclude that they were not mature cells but MBP is not expressed in the cell soma of mature cells. These conclusions should either be reworded or more appropriate immunostainings should be conducted, perhaps CNP?

Reviewer #1 (Remarks to the Author):

In the manuscript with the title « Myelination of parvalbumin interneurons shapes the function of cortical sensory inhibitory circuits, the team of Maria-Cecilia Angulo demonstrated that genetic disruption of GABAergic synaptic signalling between fast-spiking interneurons (FSI) and oligodendrocyte precursor cells results in severe FSI myelination defects later in development. FSI with altered axonal myelination showed reduction in high-frequency firing and weaker connectivity with excitatory cortical neurons. Feedforward inhibition performed by FSI was also reduced. These alterations were accompanied by diminished whisker-dependent texture discrimination in the behavioural tests.

In the revised version of their manuscript, the authors addressed the comments of the reviewers by performing additional experiments, including new pieces of data, re-arranging the text of the manuscript, and improving the Discussion section. I think that the manuscript has become clearer and more complete now, and appears stronger than the original version.

The major strengths of this study are the novelty and the combination of multiple state-of-the-art approaches (including electrophysiology, imaging, modelling, behavioural paradigms, and transgenic animals) used to obtain the results. The findings of this study are clearly novel, and are expected to be of interest for neuronal and glial physiologists, as well as for clinical scientists. Neuronal physiologists rarely consider functional significance of myelination for proper function of neuronal circuits, and the present study strongly emphasizes this important point. Glial physiologists are still confused about possible functional role of axon-glia synapses, and the present study shows that GABAergic axon-glia signalling is involved in regulation of myelination of interneurons. Finally, in the recent years new evidence regarding the role of myelination in schizophrenia has been emerging, and the findings of the present study suggest that disturbance of GABAergic axon-glia signalling may be a key mechanism to consider when studying myelination defects in psychiatric (and perhaps other neurological) disorders. With this in mind, it is very likely that the study of Benamer and colleagues will be highly cited and will pave the way to new future discoveries in the field.

We thank the reviewer for his/her constructive and interesting comments on our study.

Reviewer #2 (Remarks to the Author):

The authors have thoroughly addressed my concerns with the addition of new data and discussion of their results. Their manuscript will push forward the field's understanding of neuron-OPC interactions and how they influence the functional maturation of fast spiking interneurons in cortical circuits.

We thank the reviewer for his/her constructive and interesting comments on our study.

Reviewer #3 (Remarks to the Author):

Overall the authors have addressed many of my previous points however one remaining concern is the analysis of the oligodendrocyte morphology in Figure 3. The authors use CC1 staining and electrophysiology to state that the effect is exclusive to premyelinating cells. As stated before the images of the biocytin filled cells look very reminiscent of different stages of differentiation. Since CC1 is expressed early on during differentiation and continues even in mature cells I don't feel that the authors can conclusively state that premyelinating oligodendrocytes have an aberrant morphology. In my opinion one could just as easily conclude that the cells are in a different stage

of differentiation so there might be a shift in the predominant population in the f/f mice. The authors used MBP to conclude that they were not mature cells but MBP is not expressed in the cell soma of mature cells. These conclusions should either be reworded or more appropriate immunostainings should be conducted, perhaps CNP?

We thank the reviewer for his/her last question that we now addressed in the revised manuscript. First, we replaced the term « premyelinating OL » by « differentiating OL » throughout the text because the latter does not refer to cells in a specific differentiation state. Then, we suppressed the two sentences stating that these cells had an aberrant morphology (Page 10 and Page 19) and added that the observed morphological differences of CC1+ cells between control and mutant mice suggest that “these cells might be in a distinct differentiation state” (Page 9, bottom). Finally, our description of MBP staining in OL was probably unclear since we checked its expression in OL cell branches, but not in the soma as suggested by the reviewer. To clarify this point, we now illustrate MBP-negative branches of differentiating OL in Fig. 3c and 3d. We also included the NG2-negative staining of CC1+ OL in Fig. 3a,b and the CC1-negative staining of NG2+ cells (supplementary fig. 5a, b).